



**Distinguishing between old and modern permafrost sources with compound-**
**specific δ2H analysis**
Jorien E. Vonk[1], Tommaso Tesi[2,3], Lisa Bröder[2,4], Henry Holmstrand[2,4], Gustaf
Hugelius[4,5], August Andersson[2,4], Oleg Dudarev[6,7], Igor Semiletov[6,7,8], Örjan
Gustafsson[2,4]
[1] Department of Earth Sciences, VU University, The Netherlands
[2] Department of Environmental Science and Analytical Chemistry, Stockholm
University, Sweden
[3] ISMAR Institute of Marine Sciences, Bologna, Italy
[4] Bolin Centre for Climate Research, Stockholm University, Sweden
[5] Department of Physical Geography, Stockholm University, Sweden
[6] Pacific Oceanological Institute FEBRAS, Vladivostok, Russia
[7] Tomsk Polytechnic University, Tomsk, Russia
[8] University of Alaska Fairbanks, Fairbanks, USA
Correspondence to: Jorien Vonk (j.e.vonk@vu.nl)

**Keywords:** deuterium isotopes, yedoma, ice complex deposit, n-alkanoic acids, n-
alkanes, organic matter, stable carbon isotopes, radiocarbon, Siberian Arctic,
sediments, permafrost thaw
**Abstract**
Pleistocene ice complex permafrost deposits contain roughly a quarter of the organic
carbon (OC) stored in permafrost terrain. When permafrost thaws, its OC is
remobilized into the (aquatic) environment where it is available for degradation,
transport or burial. Aquatic or coastal environments contain sedimentary reservoirs
that can serve as archives of past climatic change. As permafrost thaw is increasing
throughout the Arctic, these reservoirs are important locations to assess the fate of
remobilized permafrost OC.
We here present compound-specific deuterium ($\delta^2$H) analysis on leaf waxes as a tool
to distinguish between OC released from thawing Pleistocene permafrost (Ice
Complex Deposits; ICD) and from thawing Holocene permafrost (from near-surface
soils). Bulk geochemistry (%OC, $\delta^{13}$C, %total nitrogen; TN) was analyzed as well as
the concentrations and $\delta^2$H signatures of long-chain $n$-alkanes ($C_{21}$ to $C_{33}$) and
mid/long-chain $n$-alkanoic acids ($C_{16}$ to $C_{30}$) extracted from both ICD-PF samples
(n=9) and modern vegetation/O-horizon (Topsoil-PF) samples (n=9) from across the
northeast Siberian Arctic.
Results show that these Topsoil-PF samples have higher %OC, higher OC/TN values,
and more depleted $\delta^{13}$C-OC values than ICD-PF samples, suggesting that these former
samples trace a fresher soil and/or vegetation source. Median concentrations of high-
molecular weight $n$-alkanes (sum of $C_{25}$-$C_{27}$-$C_{29}$-$C_{31}$) were 210±350 µg/gOC
(median±IQR) for Topsoil-PF and 250±81 µg/gOC for ICD-PF samples. Long-chain $n$-



alkanoic acids (sum of $C_{22}$-$C_{24}$-$C_{26}$-$C_{28}$) were more abundant than long-chain $n$-
alkanes, both in Topsoil-PF samples (4700±3400 µg/gOC) and in ICD samples
(6630±3500 µg/gOC). Whereas the two investigated sources differ on the bulk
geochemical level, they are, however, virtually indistinguishable when using leaf wax
concentrations and ratios.
However, on the molecular-isotope level, leaf wax biomarker $\delta^2H$ values are
statistically different between Topsoil-PF and ICD-PF. The mean $\delta^2H$ value of $C_{29}$ $n$-
alkane was -246±13‰ (mean±stdev) for Topsoil-PF and -280±12‰ for ICD-PF,
whereas the $C_{31}$ $n$-alkane was -247±23‰ for Topsoil-PF and -297±15‰ for ICD-PF.
The $C_{28}$ $n$-alkanoic acid $\delta^2H$ value was -220±15‰ for Topsoil-PF and -267±16‰ for
ICD-PF. With a dynamic isotopic range (difference between two sources) of 34 to
50‰, the isotopic fingerprints of individual, abundant, biomarker molecules from
leaf waxes can thus serve as end-members to distinguish between these two sources.
We tested this molecular $\delta^2H$ tracer along with another source-distinguishing
approach, dual-carbon ($\delta^{13}C$-$\Delta^{14}C$) isotope composition of bulk OC, for a surface
sediment transect in the Laptev Sea. Results show that general offshore patterns
along the shelf-slope transect are similar, but the source apportionment between the
approaches vary, which may highlight the advantages of either. The $\delta^2H$ molecular
approach has the advantage that it circumvents uncertainties related to a marine end-
member, yet the $\delta^{13}C$-$\Delta^{14}C$ approach has the advantage that it represents the bulk OC
fraction thereby avoiding issues related to the molecular-bulk upscaling challenge.
This study indicates that the application of $\delta^2H$ leaf wax values has potential to serve
as a complementary quantitative measure of the source and differential fate of OC
thawed out from different permafrost compartments.



## 1    Introduction

Climate warming is causing permafrost soils to thaw, exposing its organic matter (OM) to decomposition (e.g., Schuur et al., 2015; Zimov et al., 1993; Semiletov et al., 2012). Thaw will increase the hydrological connectivity of landscapes and will cause release of OM into the aquatic environment (Walvoord et al., 2012; Vonk et al., 2015; Anderson et al., 2011). Here, the OM can continue to decompose, generating greenhouse gases (e.g., Semiletov et al., 1996a,b; Anderson et al., 2009; Shakhova et al., 2015), or be destined for burial into inland and coastal sediments. These sedimentary archives serve as long- and short-term reservoirs that attenuate greenhouse gas emissions from thawing permafrost (Vonk and Gustafsson, 2013; Semiletov et al., 2011).

The release of OM from thawing permafrost into aquatic sediments varies over time and space. A recent study showed that at the end of the last glacial, the surface active layer of terrestrial permafrost released about 4.5 Tg organic carbon (OC) per year from just the Lena watershed onto the nearby shelf, whereas current annual OC release is estimated to be only about a tenth of this (Tesi et al., 2016). In addition to active layer material, OM from deeper and older permafrost sources can also thaw and be released into the environment (Shakhova et al., 2007, 2014). This process currently dominates the delivery of terrestrial material onto the East Siberian Arctic shelf (Vonk et al., 2012; Semiletov et al., 1999) and is expected to increase due to accelerating coastal erosion rates (Günther et al., 2013).

Different permafrost OC stocks exhibit variable vulnerabilities to thaw remobilization (Schuur et al., 2015). In addition to a subsea permafrost OC stock, soils and sediments of the terrestrial northern permafrost zone store about 1300±200 Pg OC, with separate upscaling approaches applied for soil stocks (0-3m depth), deltaic sediments (full depth) and Yedoma sediments (full depth) (Hugelius et al., 2014). Yedoma sediments, a.k.a. Ice Complex Deposits (ICD) are polygenetic, ice-rich Pleistocene-aged deposits that are present in the unglaciated parts of Siberia and Alaska (Schirrmeister et al., 2011). These deposits contain roughly a quarter of the OC stored in permafrost terrain, but estimates vary from ca. 200-400 Pg C (Strauss et al., 2013; Schuur et al., 2015). The presence of massive ice wedges in ICD causes landscapes to collapse upon thaw, exposing deeper stocks of OC. This type of relatively abrupt thaw is increasing in many parts of the arctic landscape (Schuur et al., 2015). At the same time, deepening of the active layer causes gradual thaw that occurs across entire landscapes (Shiklomanov et al., 2013).

With a tool to detect and monitor different types of permafrost OM in coastal environments, one could assess (historical and spatial) variability in permafrost source input, degradation and thaw, as well as the relative degradation of different permafrost types. For example, the relative release of OC from ICD versus topsoil permafrost has earlier been distinguished and quantified through the use of dual-carbon isotopes ($\delta^{13}C$ and $\Delta^{14}C$) on bulk OC in the shelf environment of the Laptev and East Siberian Sea. It was shown that topsoil permafrost OC dominates in suspended particulate matter (Karlsson et al., 2011; 2016; Vonk et al., 2012) and ICD





permafrost OC dominates in the surface sediments (Vonk et al., 2012; Semiletov et al.,
2011; 2012). Vonk et al. (2014) further showed that topsoil OC is actively degraded
during horizontal transport whereas ICD permafrost OC rapidly settles. Winterfeld et
al. (2015) showed, using dual-carbon isotopes on riverine material, that suspended
particulate OC in the Lena Delta mostly consists of Holocene material instead of
material from ICD permafrost.
This $\delta^{13}C$-$\Delta^{14}C$ dual-carbon isotope approach carries the strong advantage that it
operates on the bulk OC level, thereby circumventing the issues associated with
molecular isotope proxies that relate to upscaling from the molecular to the bulk level
(e.g. selective degradation, differences in physical association, dispersion
differences). However, the $\delta^{13}C$-$\Delta^{14}C$ approach also has drawbacks, such as a weak
distinction between the $\delta^{13}C$ end-member values of Topsoil-PF versus ICD-PF. Also,
the marine $\delta^{13}C$ end member values in coastal Arctic shelf waters are uncertain and
may be more depleted than at mid-latitudes due to uptake of relatively depleted
dissolved $CO_2$ values caused by cold polar water (Meyers, 1997; Tesi et al. *this special*
*issue*) or degradation of terrestrial matter (Anderson et al., 2009; 2011; Semiletov et
al., 2013; 2016), generating a potential overlap between marine and topsoil $\delta^{13}C$ end-
members.
Here we propose a complementary tool to trace permafrost OC release into the
coastal environment based on molecular $\delta^2H$ analysis on leaf waxes. Isotopes in water
molecules ($\delta^2H$ or $\delta^{18}O$) in glacial ice cores as well as in massive ground ice in the
northern hemisphere have been used for reconstructing palaeotemperatures (Kotler
and Burn, 2000; Johnson et al., 2001; Meyer et al., 2015) as the isotopic value of local
precipitation is a function of local climate (Sachse et al., 2004; Smith and Freeman,
2006). Higher plants use water as their primary source of hydrogen during
photosynthesis (Sternberg, 1988). The $\delta^2H$ isotope values of leaf wax *n*-alkanoic acids
or *n*-alkanes are therefore reflecting the $\delta^2H$ isotopic value of local precipitation (e.g.,
Sachse et al., 2004; Sessions et al., 1999), after correction for the net fractionation
during biosynthesis, and evapotranspiration (Leaney et al., 1985). Global
precipitation values can vary by as much as 200‰ with values around 0‰ in the
tropics but approaching -200‰ near the North Pole (www.iaea.org). Additionally,
the fractionation between source water and plant wax molecules varies both in time
and space, and can be up to -170‰ (Smith and Freeman, 2006; Sachse et al., 2004;
Polissar and Freeman, 2010) but appears relatively small at higher latitudes
(between -59 and -96‰; Shanahan et al., 2013; Wilkie et al., 2013; Porter et al., 2016).
Differences in $\delta^2H$ signatures of leaf wax molecules from terrestrial regions with
different (past) climates could therefore potentially be applied to derive the relative
proportion of different types of thawing permafrost in nearby coastal settings. We
hypothesize that $\delta^2H$ signatures of leaf wax *n*-alkanoic acids and *n*-alkanes are more
depleted in OC from permafrost deposits formed during the colder and drier
Pleistocene, compared to more enriched values in OC from active layer or surface
permafrost formed during the warmer Holocene.



This study investigates a source-specific $\delta^2H$ signature for both ICD permafrost and
recent, surface soil permafrost in Northeast Siberia. Furthermore, we explore the
possibilities of using these isotopic end-member values in regional source-
apportionment calculations that aim to quantify the relative contribution of different
sources of permafrost OC. As permafrost thaw progresses, particularly in ice-rich
permafrost such as ICD, it is increasingly important to trace the fate of remobilized
and decomposing OC in the Arctic environment.
**2      Methods**
**2.1     Sampling**
A total of 18 samples were collected throughout the Siberian Arctic. Recent surface
soils (n=7) and vegetation (n=2) samples were analyzed and (from here on) referred
to as the "topsoil" permafrost (Topsoil-PF) sample set, whereas ICD-PF samples were
obtained from ICD soil profiles (n=7) and suspended particulates from ICD
formations (n=2) (Fig. 1 and Table 1). Eight offshore sediments along a shelf-slope-
continental rise transect in the Laptev Sea were collected in 2014, further marine
sampling details can be found in Bröder et al. (2016b).
The Topsoil-PF samples represent O and A soil genetic horizons in sites with active
soil formation. The sites where chosen to represent typical soil and vegetation types
in the investigated permafrost landscapes, including both taiga and tundra sites.
Samples were collected by depth or soil horizon increments from open soil pits using
fixed volume sampling procedures.
The ICD-PF samples were collected from vertical exposures that were excavated to
expose intact permafrost. Fixed-volume samples were collected by coring
horizontally into the frozen sediments to extract ICD-PF samples from consecutive
depths.
For more details about sampling sites, including location, vegetation and soil types
see table 1 (terminology following the U.S.D.A. Soil Taxonomy; Soil Survey Staff,
2014). Sampling was done in late summer near the time of maximum annual active
layer depth, in July 2010 (CH DY-3A and 4A; Vonk et al. (2013)) and August 2011
(Palmtag et al., 2015) for the Kolyma River region, in August 2012 for the lower Lena
River and Indigirka River (Siewert et al., 2015; Weiss et al., 2015) and in August 2013
for the upper Lena River (Siewert et al., 2016).  For more detailed descriptions of
sample collection we refer to these references.  The vegetation samples CH Medv
grass and CH Y4 grass were obtained from the tundra near Medvezhka River and a
birch forest near Y4 stream, respectively, in July 2012.
Samples CH DY-3A and 4A were obtained in July 2010 at the Duvannyi Yar ICD
exposure along the Kolyma River (Vonk et al., 2013). The particulate sediment
samples were taken from thaw streams that were freshly formed from thawing ICD
(transport time from thaw to sampling estimated to be less than 1h).





## 2.2 Analytical methods

Freeze-dried samples were extracted using an ASE 200 accelerated solvent extractor (Dionex Corporation, USA) using DCM/MeOH (9:1 v/v) at 80°C ($5 \times 10^6$ Pa) (Wiesenberg et al., 2004). After the extraction, solvent-rinsed activated copper and anhydrous sodium sulfate were added to the extracts to remove sulfur and excess water, respectively. After 24 h, extracts were filtered on pre-combusted glass wool and concentrated with the rotary evaporator. Extracts were transferred into glass tubes, evaporated to complete dryness and re-dissolved in 500 µl of DCM. Lipid fractionation was performed via column chromatography using amino-propyl Bond Elut (500 mg/3 ml) to retain the acid fraction and $Al_2O_3$ to separate the hydrocarbon and polar fractions (Vonk et al., 2010).

Prior to the analyses, saturated *n*-alkanes (hydrocarbon fraction) were further purified using 10% $AgNO_3$ coated silica gel to retain the unsaturated fraction. The acid fraction was methylated using a mixture of HCl, MilliQ water and methanol at 80°C overnight to obtain the fatty acid methyl ester (FAME) fraction. Methylated acids were extracted with hexane and further purified using 10% $AgNO_3$ coated silica gel. The hydrocarbon and FAME fractions were quantified via gas chromatography mass spectrometry (GC–MS) in full scan mode (50-650 m/z) using the response factors of commercially available standards (Sigma-Aldrich). The GC was equipped with a 30 m×250 µm DB5-ms (0.25 µm thick film) capillary GC column. Initial GC oven temperature was set at 60°C followed by a 10°C min$^{-1}$ ramp until a final temperature of 310°C (hold time 10 min).

The hydrogen-isotopic composition of hydrocarbon and FAME fractions was measured with continuous-flow GC - isotope ratio - MS. Purified extracts were concentrated and injected (1-2 µl) into a Thermo Trace Ultra GC equipped with a 30m×250 µm HP5 (0.25 µm thick film) capillary GC column. Oven conditions were similar to the setting used for the quantification. The conversion of organic biomarkers to elemental hydrogen was accomplished by high-temperature conversion (HTC) at 1420°C (Thermo GC Isolink). After the HTC, $H_2$ was introduced into the isotope ratio MS (Thermo Scientific™ Delta V™IRMS) for compound-specific determination of $\delta^2H$ values via a Thermo Conflo IV. Following a linearity test, we only used peaks with amplitude (mass 2) between 1500 and 8000 mV for the evaluation. The $\delta^2H$ values were calibrated against saturated HMW *n*-alkanes using the reference substance mix A4 (Biogeochemical Laboratories, Indiana University). The H3+ factor was determined every day and stayed constant (<3) throughout the evaluation. Each purified extract was injected three times. FAMEs were further corrected to account for the methylation agent by comparing the hydrogen abundance of lauric acid ($C_{12}$-FA; i.e. 12 carbon atoms) as acid and corresponding methyl ester. The average methylation effect for lauric acid was 23.97±3.9‰ (n=4). This factor was, normalized to chain length (i.e. increasing chain lengths result in lower corrections), applied to all the FAMEs. $\delta^2H$ values of *n*-alkanes and FAMEs are reported as mean, standard deviation and weighted average (Table 5).



### 2.3 Source apportionment

The compound-specific $\delta^2$H signatures in this study were used to differentiate between the two major sources (end-members), Topsoil-PF and ICD-PF, using an isotopic mass-balance model. We used a Markov chain Monte Carlo (MCMC) approach to account for the end-member variability (Andersson et al., 2015; Bosch et al., 2015). The end-members were represented by normal distributions, with mean and standard deviations obtained from the literature values. For each Laptev Sea station, the isotope signatures from three different terrestrial molecular markers (long-chain $n$-alkanes $C_{27}$, $C_{29}$ and $C_{31}$) were used jointly to improve source apportionment precision. The $\delta^2$H signatures for the two end-members were based on our Topsoil-PF and ICD-PF samples (see Table 5 for mean source values).

The compound-specific $\delta^2$H-based source apportionment was compared to $\Delta^{14}$C/$\delta^{13}$C-based analysis of bulk OC using analogous MCMC techniques (e.g., Vonk et al., 2012). The $\Delta^{14}$C/$\delta^{13}$C-approach allows estimation of the relative contribution of a third source, marine, which does not affect the presently investigated (terrestrial) compounds. Accounting for the marine component to OC allows direct comparison of the Holocene and Pleistocene contributions. All MCMC calculations were made using Matlab scripts (ver. 2014b) using 200,000 iterations, a burn-in phase (initial search period) of 10,000 and a data thinning of 10.

The spatial extent of ICD in the Lena River Basin was calculated by overlaying the extent of the drainage basin (from WRIBASIN: Watersheds of the World published by the World Resources Institute, www.wri.org/publication/watersheds-world) with the extent of the Yedoma Region (digitized from Romanovsky, 1993) in an equal area map projection. It was assumed that 30% of the Yedoma Region consists of intact ICD (following Strauss et al., 2013).

## 3 Results

### 3.1 Bulk geochemistry

The investigated Topsoil-PF and ICD-PF samples are, on a bulk geochemical level, very different. Mean organic carbon contents (as %OC) and total nitrogen content (as %TN) are 25±12 and 1.1±0.67 for Topsoil-PF samples, and 1.6±0.31 and 0.17±0.058 for ICD-PF samples, respectively (Table 1). This gives TOC/TN ratios of 25±8.0 for Topsoil-PF samples and 10±2.6 for ICD-PF samples. Stable carbon isotopic values of Topsoil-PF and ICD-PF samples are -27.8±1.3‰ and -25.7±0.75‰, respectively (Table 1).

### 3.2 Molecular geochemical composition

Long-chain $n$-alkanes and $n$-alkanoic acids are abundant in epicuticular waxes and therefore indicative for a source of higher plants (Eglinton and Hamilton, 1967). Concentrations of individual long-chain $n$-alkanes in Topsoil-PF samples ranged from 1 to 340 µg/gOC ($C_{21}$-$C_{33}$; Table 2) with an average chain length of 28±1.6. The sum of high-molecular weight (HMW) $n$-alkanes (>$C_{21}$) was 420±330 µg/gOC





(median±IQR; interquartile range) and the most abundant $n$-alkanes added up to
210±350 µg/gOC (sum of $C_{25}$-$C_{27}$-$C_{29}$-$C_{31}$) (Table 4, Fig. 2a). For ICD-PF samples, the
individual concentrations of long-chain $n$-alkanes were between 4 and 160 µg/gOC,
and the average chain length 27±0.7 (Table 2). The sum of high-molecular weight $n$-
alkanes, and most abundant $n$-alkanes were 700±180 µg/gOC and 350±81 µg/gOC,
respectively (Table 4, Fig. 2a). The carbon preference index (CPI), a molecular ratio
indicative for degradation status with values >5 typical for fresher terrestrial material
and values approaching 1 typical for more degraded samples (Hedges and Prahl,
1993), showed values for Topsoil-PF samples of 7.3±3.6 (average±standard
deviation) and ICD-PF samples of 3.6±0.8 (CPI $C_{23}$-$C_{31}$; Table 4, Fig. 2c). The
$C_{25}/(C_{25}+C_{29})$ ratio, indicative for the input of peat moss (*Sphagnum sp.*) material
(Vonk and Gustafsson, 2009; *Sphagnum* values 0.72, higher plants 0.07; Nott et al.,
2000) was 0.33±0.22 (average±standard deviation) and 0.34±0.05 for Topsoil-PF and
ICD-PF samples, respectively (Table 4).
Long-chain $n$-alkanoic acids ($C_{22}$ and above) were abundant in concentrations
between 0.122 and 2670 µg/gOC for individual homologues in topsoils, with the sum
of HMW $n$-alkanoic acids (>$C_{22}$) being 6400±4300 µg/gOC (median±IQR) and the
most abundant $n$-alkanoic acids (sum of $C_{22}$-$C_{24}$-$C_{26}$-$C_{28}$) adding up to 4700±3400
µg/gOC (Table 2, 4 and Fig. 2b). ICD-PF samples contained individual long-chain $n$-
alkanoic acids in 2.17 and 18700 µg/gOC (Table 3), a sum of HMW $n$-alkanoic acids of
8300±5100 µg/gOC, and the sum of most abundant, even $n$-alkanoic acids of
6600±3500 µg/gOC (Table 4). Topsoil-PF and ICD-PF samples had average chain
lengths of 24.1±1.1 and 24.3±0.59, and CPI ($C_{22}$-$C_{28}$) values of 5.9±2.7
(average±standard deviation) and 5.0±1.6, respectively (Table 4). Shorter-chain $n$-
alkanoic acids $C_{16}$ and $C_{18}$ are produced in basically all types of life in soils or aquatic
environments, and are not specific for higher plants. Topsoil-PF contained $C_{16}$ and $C_{18}$
homologues in concentrations between 220 and 4600 µg/gOC, and ICD-PF samples
between 200 and 10400 µg/gOC (Table 3).
Degradation of organic matter involves the loss of functional groups, e.g. the loss of
carboxylic acids (Meyers and Ishiwatari, 1993). A high ratio of HMW $n$-alkanoic acids
over HMW $n$-alkanes in a sample therefore implies a relatively fresh, less degraded,
status (i.e. relatively more functional groups present). For Topsoil-PF samples, the
HMW $n$-alkanoic acid/HMW $n$-alkane ratio varied between 5.6 and 25 with an
average value of 13±7.6, whereas ICD-PF samples varied between 7.6 and 140 with
an average value of 29±43 (Table 4, Fig. 2f).

### 3.3    Molecular isotopic composition

We measured $\delta^2H$ values in long-chain $n$-alkanes and $n$-alkanoic acids between -119
and -313‰ (Fig. 3, Table 5). Mean values for HMW $n$-alkanes ($C_{25}$-$C_{27}$-$C_{29}$-$C_{31}$) were
between -201 and -247‰ for Topsoil-PF samples and between -221 and -297‰ for
ICD-PF samples, with consistently lower $\delta^2H$ for longer chain lengths. For HMW $n$-
alkanoic acids ($C_{22}$-$C_{24}$-$C_{26}$-$C_{28}$) mean $\delta^2H$ values were between -203 and -236‰ for
Topsoil-PF samples and between -261 and -278‰ for ICD-PF samples (Table 5). The





decrease in $\delta^2H$ values with increasing chain length is less distinct for *n*-alkanoic acids
but one can observe a decrease of around 25-30‰ from $C_{22}$ to $C_{26}$ (Fig. 3). For ICD-
PF samples, it seems that the isotopic depletion for the average of the three most
abundant *n*-alkanes is comparable to the average for *n*-alkanoic acids, whereas in
Topsoil-PF samples, the isotopic depletion for the three most abundant *n*-alkanes is a
bit larger than for *n*-alkanoic acids (Fig. 4).

## 4       Discussion

### 4.1       Using bulk geochemistry and molecular proxies

Bulk geochemical and isotopic analysis, as well as analysis of molecular proxies
remained inconclusive in distinghuishing between the two investigated sources in
this study. Topsoil-PF samples have a higher organic content, higher TOC/TN values
(representing fresh, higher plant material; Meyers, 1994) and more depleted $\delta^{13}C$
values (indicative for terrestrial C3 plants; Meyers, 1997) than ICD-PF samples,
suggesting that these samples indeed trace a fresh soil and/or vegetation source
(Table 1). The $\delta^{13}C$ values of a larger ICD-PF and Topsoil-PF dataset have earlier been
summarized (Vonk et al., 2012 and references therein; Schirrmeister et al., 2011)
giving values of -26.3±0.67‰ (n=374) and -28.2±2.0‰ (n=30), respectively. Our
values (Table 1) are in a similar range. Despite the differences between these two
sources in their bulk geochemistry, it is hard to use these parameters for source
distinction as their variability is fairly high, and their behavior in the environment is
not conservative, but e.g. affected by degradation processes. On a molecular
geochemical level the two investigated sources are virtually indistinguishable as
there is a considerable variation in molecular concentrations and proxy values (Fig.
2). Only one of the tested parameters, the CPI $C_{23}$-$C_{31}$ of *n*-alkanes, showed a
statistically significantly different value for the two investigated sources.

### 4.2       Evaluation of molecular $\delta^2H$ values as a source end-member

To alleviate the difficulty to distinguish between Topsoil-PF and ICD-PF with just bulk
and molecular geochemical characteristics, we explore the $\delta^2H$ values of leaf wax
molecules (i.e. long chain *n*-alkanoic acids and *n*-alkanes) to differentiate between
their relative source contributions. The overall mean $\delta^2H$ of the four most abundant
*n*-alkanoic acids is -231±29‰ and -271±13‰ for Topsoil-PF and ICD-PF samples,
respectively. These values compare well with available literature (Fig. 5). Pautler et
al. (2014) measured $\delta^2H$ values on $C_{29}$ *n*-alkanes in modern soils of the Yukon, Canada
of -252±9.1‰ (n=4) and aged soil $\delta^2H$ values of -269±8.6‰ (n=13; 24-25 [14]C-ka ago)
and -273±16.4‰ (n=9; for MIS 4, ~70 [14]C-ka ago). Yang et al. (2011) also reported
$C_{29}$ *n*-alkane $\delta^2H$ values for modern vegetation from Alaska and Arctic Canada with an
average value of -252±43‰ (n=8). Zech et al. (2011) reported values of $C_{29}$ *n*-alkanes
collected from a permafrost exposure along the Tumara River in northeast Siberia,
with an average value of -266±7.5‰ (n=23) for glacial paleosoils and -247±9.4‰
(n=17) for interglacial paleosoils. Our values for $C_{29}$ *n*-alkanes for Topsoil-PF (-
246±13‰; n=9) and ICD-PF (-280±12‰; n=9) are in a similar range (Fig. 5). For $C_{28}$
*n*-alkanoic acids, Wilkie et al. (2013) measured -252±8.7‰ (n=6) for modern





vegetation in northeast Siberia, whereas Porter et al. (2016) measured -269±2.7‰ (n=7) for ca. 31 cal ka BP old soils in the Yukon. Compared to these studies, our values for $C_{28}$ *n*-alkanoic acids are somewhat more enriched for Topsoil-PF with -220±15‰ (n=7) but roughly in the same range for ICD-PF with -267±16‰ (n=9).

The mean isotopic difference between the most abundant *n*-alkanoic acids of the two investigated sources is around 40‰ ($\delta^2H$ values of -231±29‰ and -271±13‰ for Topsoil-PF and ICD-PF samples, respectively). Despite the relatively large standard deviations, the isotopic differences are statistically significant for each of the *n*-alkanoic acids individually ($C_{22}$, $C_{24}$, $C_{26}$, $C_{28}$; Fig. 3). The isotopic differences between the two sources for the mean value of the four most abundant *n*-alkanes is 35‰, with a mean value of -229±33‰ and -264±34‰ for Topsoil-PF and ICD-PF samples, respectively. Here, the individual *n*-alkane isotopic signatures are statistically significantly different for $C_{27}$, $C_{29}$, $C_{31}$ (Fig. 3) in Topsoil-PF and ICD-PF samples. The selection and application of individual chain length $\delta^2H$ values as end-members, in contrast to mean chain length values, might be more appropriate for several reasons; (i) to reduce variability ($\delta^2H$ ranges for $C_{29}$ and $C_{31}$ *n*-alkanes and $C_{22}$ and $C_{24}$ *n*-alkanoic acids are relatively low; Fig. 3), (ii) to target the most abundant species ($C_{29}$ and $C_{31}$ *n*-alkanes are generally more abundant in soils and ICD-PF compared to shorter chain lengths; Table 2), and (iii) to make use of the largest dynamic range between source end-member values ($C_{31}$ *n*-alkane $\delta^2H$ values of Topsoil-PF and ICD-PF differ by 50‰). Based on these arguments, the $C_{28}$ *n*-alkanoic acid and the $C_{29}$ or $C_{31}$ *n*-alkanes are most appropriate to use for source-apportionment. The available previous studies (Fig. 5) have also selected these chain lengths ($C_{28}$ *n*-alkanoic acid and $C_{29}$ *n*-alkanes) for proxy development.

The use of molecular $\delta^2H$ values as tracers of terrestrial material in a marine or coastal setting has the advantage that it avoids uncertainty issues related to definition of the marine end-member. On the other hand, the inherent bulk-upscaling challenge of any molecular proxies, is a disadvantage of the $\delta^2H$ approach as it introduces unknowns related to the molecular-bulk upscaling effort (e.g. taking into account sorting and recalcitrance; discussed in depth in 4.3). We also want to emphasize that $\delta^2H$ leaf wax values in the two studied end-member sets (Topsoil-PF vs. ICD-PF) largely depend on the climate (warm vs. cold) and continentality (near the coast vs. further inland) during plant formation, and associated differences in fractionation mechanisms. Consequently, when $\delta^2H$ values in samples are used for source-apportionment, this may represent the fraction leaf wax produced in cold vs. warm conditions (as well as degree of continentality), and not necessarily the fraction Topsoil-PF vs. ICD-PF.

**4.3 Comparison with $^{13}C$-$^{14}C$ source-apportionment: a case-study**

Bulk OC dual-carbon isotope data provide a quantitative apportionment tool to assess the relative contributions of Topsoil-PF vs. ICD-PF. Here, we present a case-study of a shelf-slope transect in the Laptev Sea (Fig. 1) where both these source-apportionment tools for the first time can be applied, compared and evaluated. The





shelf-slope transect of eight surface sediment samples stretches over 600 km from the nearshore zone (72.7°N, <10m water depth) to the continental rise (78.9°N, >3000m depth) (Table 6). More molecular and bulk geochemical characteristics of these samples can be found in Bröder et al. (2016b).

The $\delta^{13}$C-$\Delta^{14}$C source-apportionment uses three end-members (marine, Topsoil-PF, and ICD-PF). End-member values are based on previously published values (Tesi et al., 2016); with a $\delta^{13}$C value of -27.0±1.2‰ (n=38; Rodionow et al., 2006; Tesi et al., 2014; Gundelwein et al., 2007; Bird et al., 2002) for Topsoil-PF, and -26.3±0.67‰ (n=374; Vonk et al., 2012; Schirrmeister et al., 2011) for ICD-PF. The Topsoil-PF $\Delta^{14}$C endmember was defined as -232±147‰ (n=29; Winterfeld et al., 2015; Jasinski et al., 1998; Kaiser et al., 2007; Höfle et al., 2013; Palmtag et al., 2015). For ICD-PF we used a $\Delta^{14}$C value of -940±84‰ (n=300; Vonk et al., 2012 and references therein). The marine end-member value was -21.0±2.6‰ (n=10; Panova et al., 2015) and -50.4±12‰ (n=10; Panova et al., 2015) for $\delta^{13}$C and $\Delta^{14}$C, respectively. Calculations were made using a Markov chain Monte Carlo approach (see 2.3).

For $\delta^2$H source-apportionment there is no need to include a marine end-member as marine organisms do not produce long-chain *n*-alkanes or *n*-alkanoic acids. We were unfortunately only able to analyze *n*-alkanes in the shelf-slope transect samples, and no *n*-alkanoic acids, due to limitations in sample volume. We used the $\delta^2$H values of the $C_{27}$, $C_{29}$ and $C_{31}$ *n*-alkanes, individually. In other words, these three chain lengths are taken as independent markers, providing an overdetermined system (i.e. two sources defined with three different markers). This is more representative than using the average (concentration-weighted) $\delta^2$H value for these *n*-alkanes as the end-member values for each chain length are different. For Topsoil-PF we used -215±39‰, -246±13‰, and -247±23‰ for $C_{27}$, $C_{29}$ and $C_{31}$ *n*-alkanes, and for ICD-PF we applied -259±18‰, -297±15‰, and -282±13‰ for $C_{27}$, $C_{29}$ and $C_{31}$ *n*-alkanes, respectively (see also Table 5). Afterwards, we averaged the three end-member contributions derived from the three calculations for each station, thereby taking the variability introduced by the end-members into account.

The source apportionment of OC from Topsoil-PF and ICD-PF to surface sediments along the Laptev Sea transect differ between the bulk $\delta^{13}$C-$\Delta^{14}$C and leaf wax $\delta^2$H approaches (Table 6). The former approach suggests Topsoil-PF contributions between 21-70%, generally decreasing offshore, and, consequently, ICD-PF contributions of 30-79%, generally increasing offshore. The latter (leaf wax $\delta^2$H) approach results in a more extreme division of sources with Topsoil-PF contributions of 83-91% and ICD-PF contributions of 9-17%, with similar patterns nearshore and offshore (Table 6). A contribution of 9-17% may seem more in line with the estimated extent of ICD in the Lena River basin: 12% of the basin falls within the Yedoma Region (as defined by Romanovsky, 1993) and about 3% consists of intact ICD (see section 2.3). However, the cross-shelf sites are also strongly influenced by coastal and/or subsea erosion (Karlsson et al. 2011; Vonk et al., 2012; Semiletov et al., 2012; 2016) so the catchment characteristics are only one part of the story. It is challenging to



interpret the differences between the two proxies but we elaborate below on
potential reasons.
Assumptions in the bulk $\delta^{13}C$-$\Delta^{14}C$ approach may affect these results. First, the
outcome of the bulk $\delta^{13}C$-$\Delta^{14}C$ approach is sensitive to the definition of the marine
end-member. Changes in the currently used $\delta^{13}C$ and $\Delta^{14}C$ value of the marine end-
member of the East Siberian Arctic Shelf (n=10; Panova et al., 2015) would likely alter
the relative Topsoil-PF and ICD-PF contributions. The currently used standard
deviation for the $\delta^{13}C$ marine end-member is 2.6‰, which is much higher than the
values for the terrestrial end-members. Second, lateral transport time enroute the
shelf-slope transect (>600 kilometers) causing potentially significant aging of
sediments and its organic carbon is not accounted for in the source-apportionment.
Lateral transport time results in older surface OC ages on the shelf, compared to those
at the initial coastal deposition. Without correcting for this factor, the source-
apportionment will generate lower contributions of the (younger) Topsoil-PF
component. In an attempt to estimate this effect, we recalculated (similar to Bröder
et al. 2016a) the relative source contributions of Topsoil-PF, ICD-PF (and marine)
with the bulk $\delta^{13}C$-$\Delta^{14}C$ approach with the assumption that the Topsoil-PF $^{14}C$ age
would be subject to a cross-shelf lateral transport time of 5000 yrs. We assumed a
linear aging along the transect based on distance from the coast, with a maximum
value of 5000 yrs aging at station SW-01. This resulted in Topsoil-PF contributions
that were up to 20% higher (for the deepest stations) compared to the source-
apportionment where lateral transport time was unaccounted for (Table 6; Fig. 6).
Assumptions in the leaf wax $\delta^{2}H$ source-apportionment approach could potentially
also impact the outcomes, and hence differences with the bulk $\delta^{13}C$-$\Delta^{14}C$ results. First,
there is an inherent assumption related to the molecular to bulk level upscaling
challenge. We assume that the physical association of $n$-alkanes in different source
end-members (Topsoil-PF vs. ICD-PF) as well as their fractionation in the coastal
system is similar. However, previous research has shown that $n$-alkanes behave
rather differently upon their release into coastal waters; $n$-alkanes originating from
surface soil or vegetation debris are not bound to minerals and remain in suspension
during transport while being actively degraded, whereas $n$-alkanes originating in
deeper mineral soils settle quickly and are protected from extensive degradation
(Vonk et al., 2010). It is possible that most of the $n$-alkanes in the Laptev Sea sediment
transect originate in (deeper) mineral soils. An effect of physical association, as well
as the potential effect of hydrodynamic sorting patterns (Tesi et al., 2016) on the leaf
wax $\delta^{2}H$ values of both sources could impact the source-apportionment. Another
factor that can introduce a bias in our leaf wax $\delta^{2}H$ approach is a proton exchange of
the C-bound H-atoms in $n$-alkanes with environmental water. Although there is no
evidence for such exchange in young (<1 million years), cold sediments (Sessions et
al., 2004) this process could be enhanced in environments of low pH. The precise
effect of such exchange on the $\delta^{2}H$ signal of our samples (or end-members) is
unknown, but we suspect this process may be minimal.



When accounting for an estimated lateral transport time, the difference in estimates of source contribution by the two different approaches (bulk $\delta^{13}C$-$\Delta^{14}C$ and leaf wax $\delta^2H$) increases offshore, from about a 25% difference near the coast to a 40% difference at stations SW-01 and SW-03. This increasing offset between the results of the two end-member mixing methods may be caused by several factors such as variability in the marine end-member (e.g. due to changes in seasonal ice cover), a selective degradation (of the topsoil OC) enroute that introduces a source bias or isotopic fractionation, or remaining factors related to the lateral transport time (incorrect assumption of 5000 years, non-linear aging along transect). These differences highlight that both source-apportionment tools still could be fine-tuned further by (i) increasing the sample size of sources to reduce end-member uncertainties, (ii) continuous adjustments in end-member values and Markov chain Monte Carlo calculations based on latest knowledge, and (iii) assuring regional testing and verification of the method when applied to new environments.

## 5     Conclusions

Leaf wax $\delta^2H$ values in samples from aquatic recipient environments can be used to source-apportion the incoming terrestrial OC into two end-members; a Pleistocene ICD permafrost source and a younger, Holocene, topsoil source. Mean isotopic values of the $C_{29}$ $n$-alkane, $C_{31}$ $n$-alkane, and $C_{28}$ $n$-alkanoic acid showed a dynamic, statistically significant range of 34, 50 and 46‰ between Topsoil-PF and ICD-PF samples, respectively, with ICD-PF samples being consistently more depleted indicative of formation during the colder and drier Pleistocene.

A case-study where we tested two isotopic proxies (leaf wax $\delta^2H$ and bulk $\delta^{13}C$-$\Delta^{14}C$) to calculate the relative terrestrial source contribution of Topsoil-PF and ICD-PF along a Laptev Sea surface sediment transect, showed that the two proxies yield variable results but overall generate similar trends offshore. We reason that variability is caused by factors such as lateral transport time, remaining uncertainties in end-member definition, or environmental factors such as physical association.

Both methods (leaf wax $\delta^2H$ and bulk $\delta^{13}C$-$\Delta^{14}C$) bring along their inherent disadvantages and advantages. The molecular approach has the distinct advantage that it circumvents the uncertainties that are associated with marine end-member definition in the case of bulk OC mixing model analysis. However, application of molecular $\delta^2H$ in source-apportionment studies brings along challenges related to the molecular-bulk upscaling step. Bulk $\delta^{13}C$-$\Delta^{14}C$ source-apportionment, on the other hand, has the advantage to operate on a bulk and perhaps more representative level, but is hampered by remaining uncertainties associated with the marine end-member.

This study shows that $\delta^2H$ of leaf wax molecules has the potential to be used in quantitative source-apportionment studies of thawing permafrost in coastal or marine settings. It can serve as an alternative or complementary approach to the commonly applied bulk $\delta^{13}C$-$\Delta^{14}C$ method. We recommend continuing optimization





of end-member definition and calibration in order to increase our understanding of
the fate of thawing permafrost in the coastal environment.
**Data availability**
All data are available in Tables 1 through 6, as well as Supplementary Table S1.
**Acknowledgements**
We would like to acknowledge Robert Spencer, Sergey Davydov, Anya Davydova,
Ekatarina Bulygina, Peter Kuhry, Matthias Siewert, Juri Palmtag, Niels Weiss, Martin
Kruså, Volker Brüchert, Pete Hill, Vladimir Mordukhovich, Alexander Charkin, Deniz
Kosmach, Per Andersson, and sampling crew and personnel of IB Oden and RV Yakob
Smirnitskyi for help with sample collection in the field. Financial support has been
provided by the Dutch NWO (Veni #863.12.004), US-NSF (Polaris Project #1044610),
the Bolin Centre for Climate Research, the Knut and Alice Wallenberg Foundation
(SWERUS-C3 Program; KAW #2011.0027), the Swedish Research Council (VR #621-
2004-4039 and 621-2007-4631), the Russian Government (mega-grant under
contract #14.Z50.31.0012 to I. S.), the Russian Science Foundation (#15-17-20032 to
O. D.), the Nordic Council of Ministers Cryosphere-Climate-Carbon Initiative (project
Defrost, #23001), the European Research Council (ERCAdG project CC-TOP #695331
to Ö.G.). This study was supported by the Delta Facility of the Faculty of Science,
Stockholm University. GH would like to acknowledge funding from ESF-CryoCarb and
EU FP7-PAGE21 projects for topsoil and ICD sample collection.
**Author contributions**
Land-based samples were collected by GH and JEV, ship-based samples were
collected by IS, OD, ÖG, TT, LB, and JEV. Laboratory analysis was performed by LB, TT,
and HH. Markov chain Monte Carlo simulations were run by AA. The manuscript was
written by JEV with input of all co-authors.

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





**Figure 1**
Map of coastal northeast Siberia showing the extent of ice complex permafrost (ICD; red)
overlaid with the location of ice complex (n=9; black diamonds) and topsoil samples
(n=9; green squares). The shelf-slope Laptev Sea transect is shown with yellow stars.

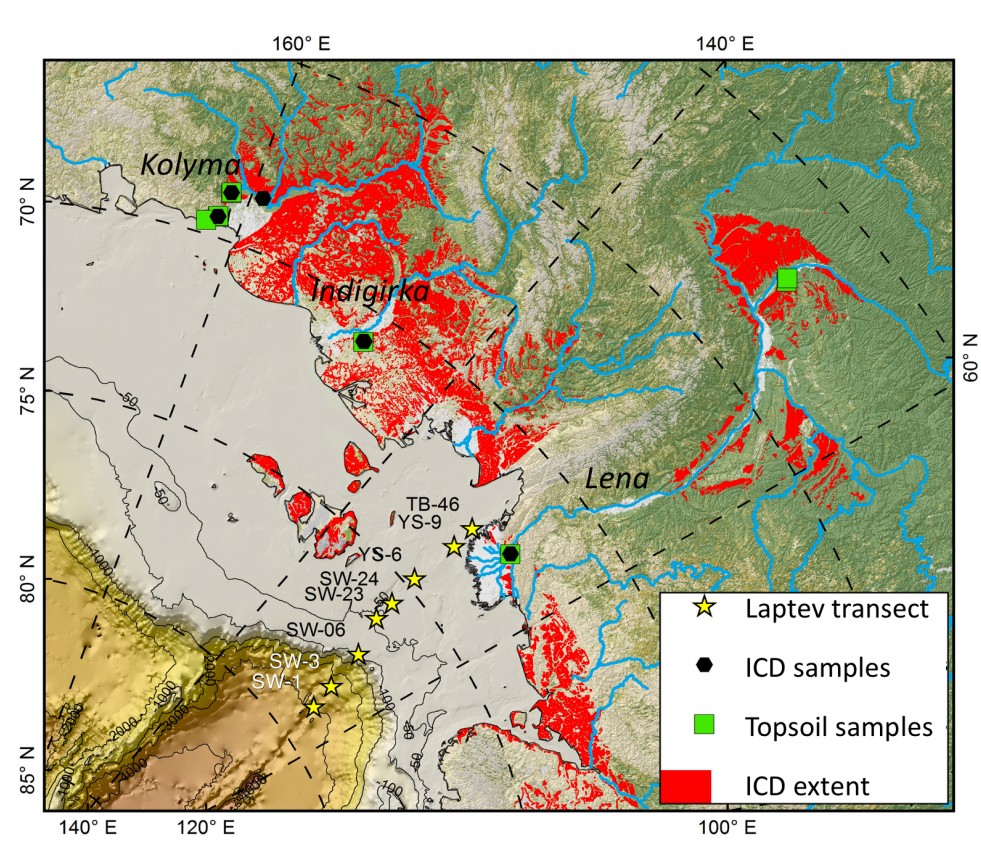






**Figure 2**
Molecular concentrations and ratios of topsoil Holocene permafrost (green) and deeper
Pleistocene permafrost (blue) samples, with (a) the sum of odd *n*-alkanes $C_{25}$-$C_{31}$, (b) the
sum of even *n*-alkanoic acids $C_{22}$-$C_{28}$, (C) the Carbon Preference Index (CPI) for *n*-
alkanes $C_{23}$-$C_{31}$, (d), the CPI for *n*-alkanoic acids $C_{22}$-$C_{28}$, (e) the ratio of $C_{25}$ over
$C_{25}$+$C_{29}$ *n*-alkanes, and (f) the sum of high-molecular weight (HMW) *n*-alkanoic acids
over HMW *n*-alkanes. The CPI is calculated as $CPI_{i-n} = \frac{1}{2} \Sigma (X_i+X_{i+2}+...+X_n)/ \Sigma (X_{i-1}+X_{i+1}+...+X_{n-1}) + \frac{1}{2} \Sigma (X_i+X_{i+2}+...+X_n)/ \Sigma (X_{i+1}+X_{i+3}+...+X_{n+1})$, where X is
concentration. Stars indicate that the two compared values are statistically significant
(95% confidence). Note that panel a and b are reported as median±IQR (interquartile
range) and the other panels are reported as average±standard deviation.

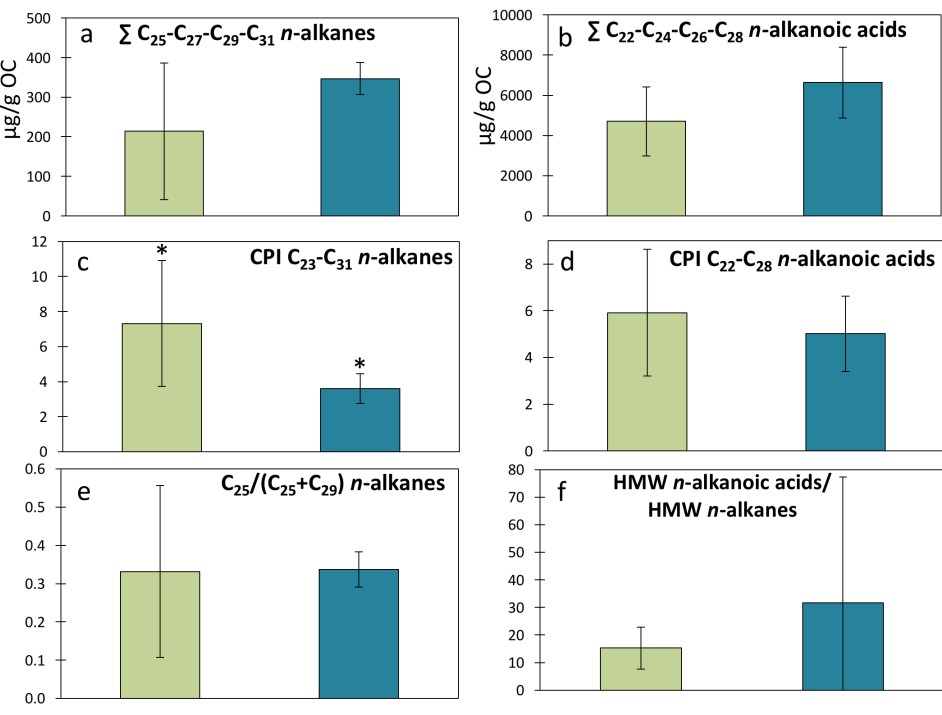



**Figure 3**
Molecular isotopic signature against chain length of long chain *n*-alkanoic acids (top) and
*n*-alkanes (bottom) for Holocene topsoil samples (green) and Pleistocene ice complex
samples (ICD; blue). Stars indicate that the two compared values are statistically
significant (95% confidence). Standard deviations are represented as vertical bars, and
are smaller than the sample circles when not visible.

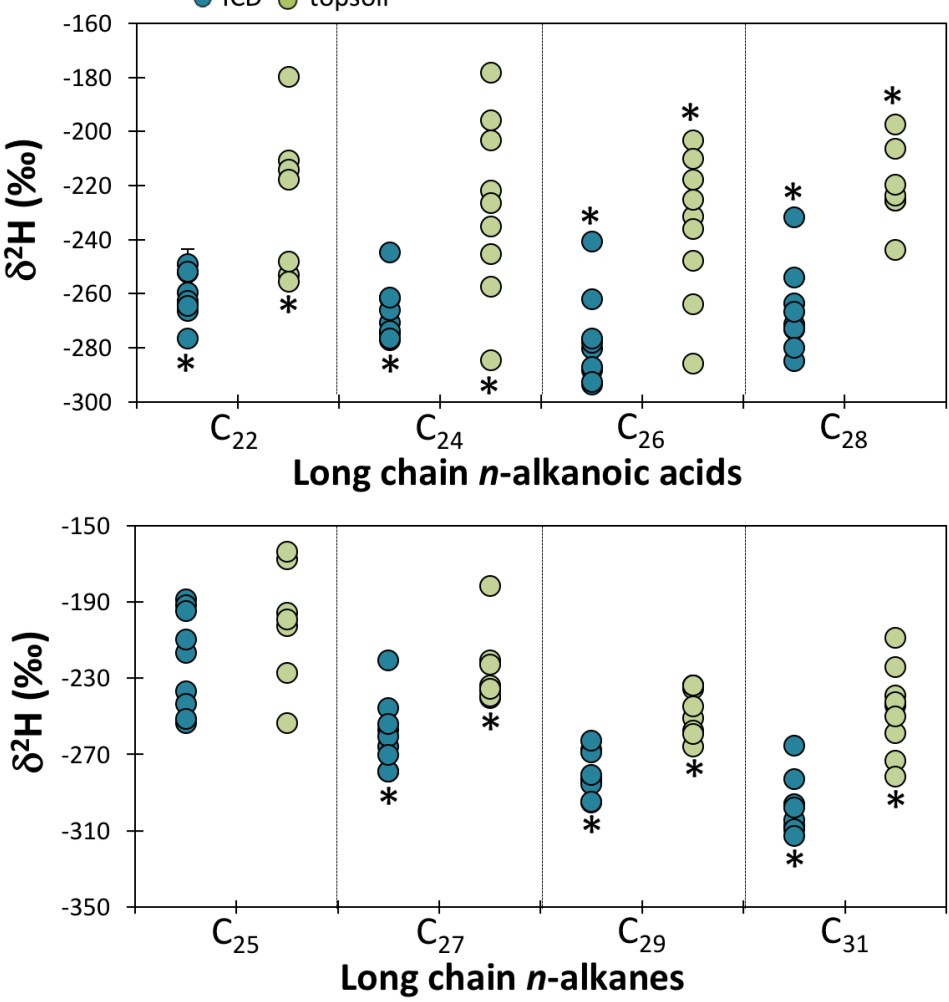





**Figure 4**

Concentration-weighted mean $\delta^2H$ values of $C_{27}$-$C_{29}$-$C_{31}$ *n*-alkanes plotted against concentration-weighted mean $\delta^2H$ values of $C_{24}$-$C_{26}$-$C_{28}$ *n*-alkanoic acids to illustrate the fractionation differences between these two leaf wax markers. Dashed line indicates an identical fractionation.

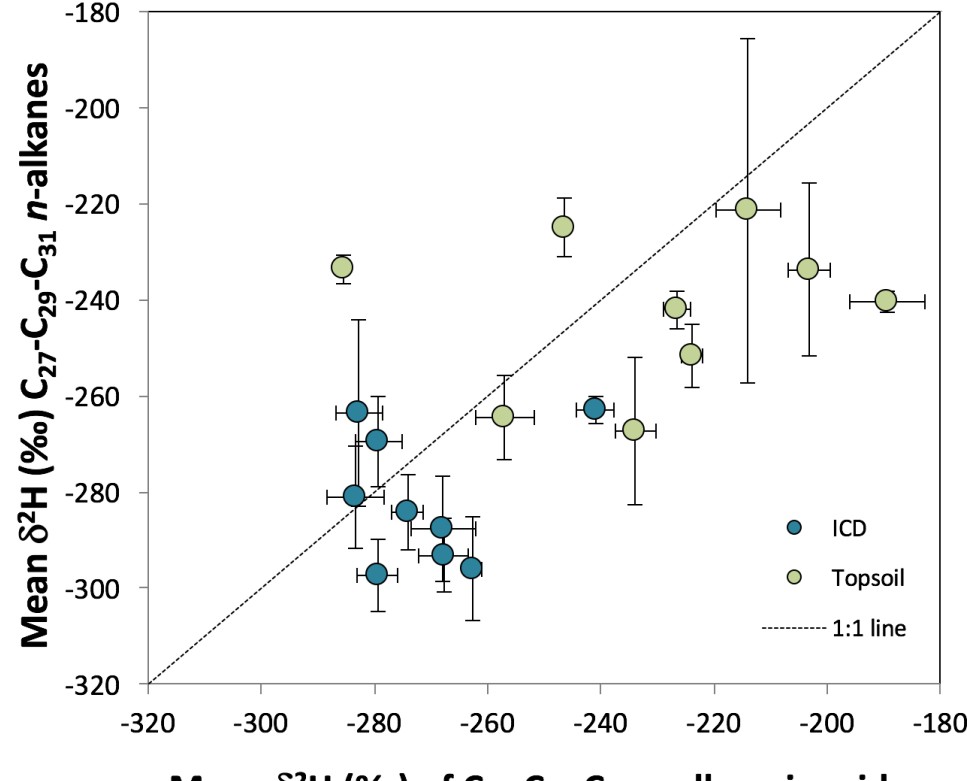



**Figure 5**

Comparison of $\delta^2H$ values of $C_{28}$ *n*-alkanoic acid (left) and $C_{29}$ *n*-alkane (right) in modern (Topsoil-PF; green circles) and ICD-PF for this study (blue circles) and available literature, with crosses from Zech et al. (2011; glacial and interglacial paleosoils from permafrost bluff exposure at Tumara River northeast Siberia), black triangles from Yang et al. (2011; C3 plants and trees from Canada and Alaska), light grey triangles from Wilkie et al. (2013; C3 plants from the El'gygytgyn lake basin, Siberia), white triangles from Pautler et al. (2014; modern and paleosoils from the Yukon territory, Canada) and dark grey triangles from Porter et al. (2016; muck deposits from the Yukon territory, Canada).

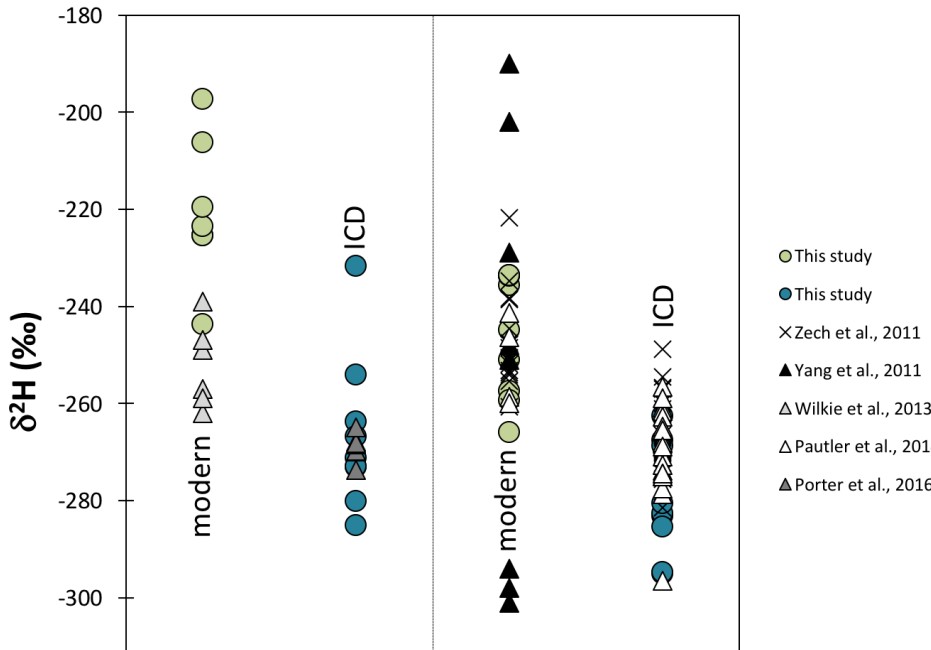





**Figure 6**
Contribution of OC from Topsoil-PF (green) and ICD-PF (blue) sources to surface
sediments along a shelf-slope transect in the Laptev Sea (see also Bröder et al., 2016b for
further transect information), calculated with a $\delta^{13}$C- $\Delta^{14}$C (triangles) and leaf wax $\delta^2$H
mixing model (circles). Stations are plotted against log water depth (m; see also Table 6)
following the transect order from the coastal, nearshore, zone in the South (furthest left;
TB-46, 6 m depth) towards the continental rise in the North (furthest right; SW-01, 3146
m depth). Topsoil $\Delta^{14}$C end-member values are corrected for cross-shelf transport time
(see section 4.2).

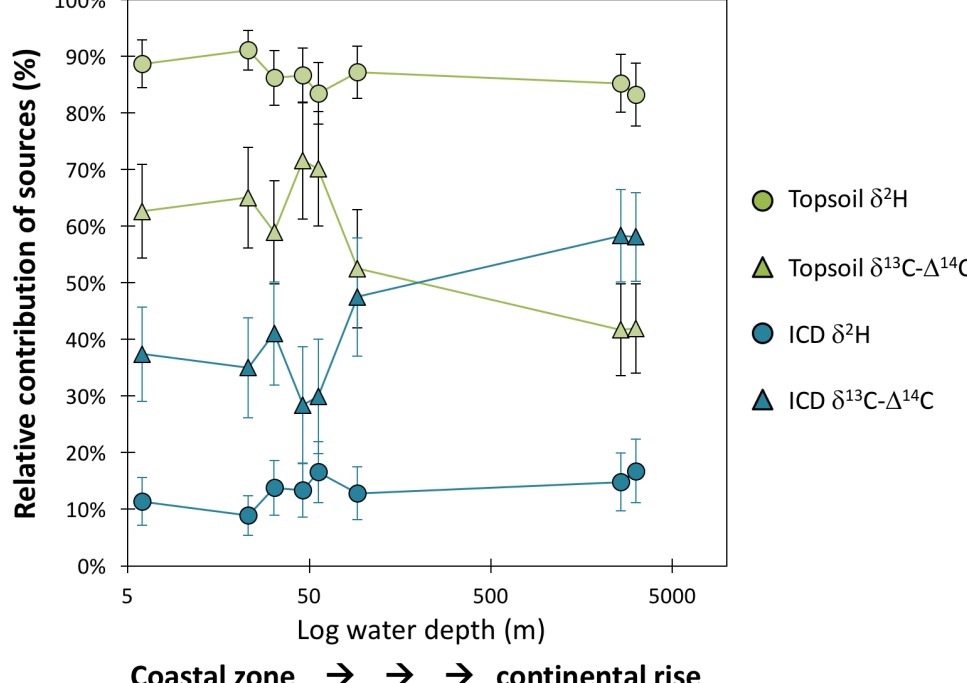





**Table 1**

Site characteristics and geochemical properties of eight topsoil and eight ice complex deposit samples. A table with more detailed sample descriptions can be found in Supplementary Table 1.

| Sample ID | Current vegetation | Watershed | Description | Lat °N | Lon °E | TOC % | $\delta^{13}C$ ‰ | TN % | C/N |
|---|---|---|---|---|---|---|---|---|---|
| *Topsoil (modern vegetation and O-horizon samples)* | | | | | | | | | |
| KU EXP 1-1, 0–16 cm | Tundra | Lena | Surface O-horizon; 0-16 cm | 72.34 | 126.29 | 11 | -27.0 | 0.40 | 27.5 |
| CH YED2, 0-4 cm | Tundra | Kolyma | Surface O-A horizon; 0-4 cm | 69.46 | 161.79 | 17 | -28.4 | 0.64 | 26.5 |
| SP T3-3B, | Alas grassland | Lena | Alas soil (Mollisol), mix of O and A horizon | 62.32 | 129.50 | 15 | -27.9 | 1.40 | 10.7 |
| SP T2-7, | Larch taiga | Lena | Taiga soil (turbel), mix of O and A horizon | 62.25 | 129.62 | 13 | -28.4 | 0.45 | 28.0 |
| KY T2-3, | Tussock tundra | Indigirka | Tundra soil (turbel), O-horizon | 70.83 | 147.48 | 29 | -28.5 | 1.56 | 18.7 |
| CH T2-1, | Tussock tundra | Kolyma | Tundra soil (turbel), mix of O, Oji and Ajj horizons | 69.44 | 161.77 | 21 | -26.4 | 0.57 | 36.7 |
| CH YED3, 0-10 cm | Larch taiga | Kolyma | Surface O-hor; 0-10 cm | 68.77 | 161.41 | 39 | -29.6 | 1.29 | 30.7 |
| CH Medv grass[a] | Grass tundra | Kolyma | Vegetation | 69.64 | 162.54 | 41 | -25.2 | 1.47 | 27.8 |
| CH Y4 grass[a] | Larch taiga | Kolyma | Vegetation | 68.74 | 161.41 | 40 | -28.5 | 2.42 | 16.6 |
| **Mean values** | | | | | | **25** | **-27.8** | **1.1** | **24.8** |
| *Ice complex deposits* | | | | | | | | | |
| KU EXP 1-3, 212-216 cm | Tundra | Lena | Very deep undisturbed yedoma ca. 10 m below surface | 72.34 | 126.29 | 1.3 | -27.5 | 0.08 | 15.7 |
| CH YED1, 300-305 cm | Tussock tundra | Kolyma | Deep undisturbed yedoma ca. 3 m below surface | 69.47 | 161.77 | 1.4 | -26.3 | 0.14 | 10.2 |
| CH YED2, 300-305 cm | Tussock tundra | Kolyma | Deep undisturbed yedoma ca. 3 m below surface | 69.46 | 161.79 | 2.3 | -25.8 | 0.27 | 8.6 |
| CH YED3, 520-525 cm | Larch taiga | Kolyma | Deep undisturbed yedoma ca. 5 m below surface | 68.77 | 161.41 | 1.4 | -25.5 | 0.15 | 9.7 |
| KY EXP1, 0-5 cm | Tussock tundra | Indigirka | Undisturbed yedoma ca. 2 m below surface | 70.83 | 147.44 | 1.5 | -25.5 | 0.18 | 8.5 |





| | | | | | | | | | |
|---|---|---|---|---|---|---|---|---|---|
| KY EXP2, 110-115 cm | Tussock tundra | Indigirka | Deep undisturbed yedoma ca. 4.5 m below surface | 70.83 | 147.44 | 1.6 | -25.6 | 0.19 | 8.6 |
| KY EXP3, 185-190 cm | Tussock tundra | Indigirka | Undisturbed yedoma ca. 2 m below surface | 70.83 | 147.49 | 1.5 | -25.2 | 0.17 | 8.5 |
| CH DY-3A | Larch taiga | Kolyma | Particulate matter from thaw streams | 68.63 | 159.15 | 1.5[b] | -25.2[b] | - | - |
| CH DY-4A | Larch taiga | Kolyma | Particulate matter from thaw streams | 68.63 | 159.15 | 1.4[b] | -25.1[b] | - | - |
| **Mean values** | | | | | | **1.6** | **-25.7** | **0.2** | **10.0** |

a vegetation samples
b data from Vonk et al., 2013



**Table 2**
Long-chain *n*-alkane concentrations (in µg/gOC) of topsoil Holocene samples (modern vegetation/O-horizon) and Pleistocene ice complex samples.

| | C21 | C22 | C23 | C24 | C25 | C26 | C27 | C28 | C29 | C30 | C31 | C32 | C33 |
|---|---|---|---|---|---|---|---|---|---|---|---|---|---|
| µg/g OC | | | | | | | | | | | | | |
| *Topsoil (modern vegetation and O-horizon samples)* | | | | | | | | | | | | | |
| KU EXP 1-1, 0-16 cm | 44 | 88 | 96 | 45 | 41 | 10 | 45 | 4.4 | 27 | 2.5 | 36 | 1.5 | 7.2 |
| CH YED2, 0-4 cm | 24 | 15 | 21 | 12 | 40 | 10 | 160 | 10 | 150 | 6.5 | 150 | 3.5 | 17 |
| SP T3-3B | 2.5 | 2.4 | 5.9 | 2.6 | 13 | 4.7 | 42 | 16 | 74 | 4.7 | 85 | 2.7 | 24 |
| SP T2-7 | 19 | 3.3 | 7.1 | 2.7 | 27 | 4.5 | 47 | 6.7 | 98 | 9.1 | 150 | 5.7 | 38 |
| KY T2-3 | 35 | 8.4 | 26 | 9.9 | 38 | 13 | 91 | 18 | 180 | 14 | 230 | 8.1 | 43 |
| CH T2-1 | 14 | 5.1 | 16 | 5.7 | 19 | 4.0 | 26 | 3.7 | 48 | 5.0 | 120 | 4.0 | 32 |
| CH YED3, 0-10 cm | 46 | 12 | 18 | 8.8 | 22 | 16 | 61 | 27 | 220 | 23 | 340 | 12 | 48 |
| CH Medv grass | 4.1 | 1.7 | 18 | 10 | 61 | 16 | 47 | 13 | 30 | 5.3 | 10 | 1.1 | 1.1 |
| CH Y4 grass | 4.7 | 2.6 | 18 | 15 | 45 | 21 | 50 | 16 | 31 | 6.8 | 9.8 | 1.5 | 2.6 |
| *Ice complex deposits* | | | | | | | | | | | | | |
| KU EXP 1-3, 212-216 cm | 57 | 79 | 100 | 49 | 82 | 23 | 170 | 16 | 137 | 8.5 | 140 | 4.4 | 25 |
| CH YED1, 300-305 cm | 55 | 89 | 100 | 70 | 70 | 27 | 75 | 20 | 130 | 12 | 120 | 5.3 | 28 |
| CH YED2, 300-305 cm | 40 | 64 | 74 | 31 | 54 | 15 | 79 | 22 | 110 | 10 | 160 | 4.8 | 32 |
| CH YED3, 520-525 cm | 60 | 93 | 98 | 47 | 55 | 20 | 84 | 22 | 140 | 12 | 150 | 6.0 | 39 |
| KY EXP1, 0-5 cm | 46 | 79 | 86 | 56 | 49 | 20 | 55 | 13 | 75 | 7.0 | 100 | 4.7 | 38 |
| KY EXP2, 110-115 cm | 41 | 73 | 87 | 68 | 62 | 29 | 65 | 20 | 98 | 11 | 120 | 4.9 | 27 |
| KY EXP3, 185-200 cm | 50 | 83 | 83 | 43 | 41 | 16 | 65 | 17 | 100 | 8.3 | 120 | 4.5 | 42 |
| CH DY-3A | 4.2 | 7.3 | 23 | 30 | 55 | 42 | 82 | 38 | 100 | 18 | 110 | 5.0 | 21 |
| CH DY-4A | 6.2 | 6.2 | 16 | 11 | 29 | 15 | 51 | 20 | 79 | 9.3 | 85 | 4.1 | 23 |





**Table 3**

Long-chain *n*-alkanoic acids concentrations (in µg/gOC) of topsoil Holocene samples (modern vegetation/O-horizon) and Pleistocene ice complex samples.

| | C16 | C18 | C20 | C21 | C22 | C23 | C24 | C25 | C26 | C27 | C28 | C29 | C30 |
|---|---|---|---|---|---|---|---|---|---|---|---|---|---|
| | µg/gOC | | | | | | | | | | | | |
| ***Topsoil (modern vegetation and O-horizon samples)*** | | | | | | | | | | | | | |
| KU EXP 1-1, 0-16 cm | 511 | 220 | 176 | 80.5 | 539 | 311 | 1100 | 4.95 | 684 | 90.5 | 350 | 32.8 | 58.1 |
| CH YED2, 0-4 cm | 1740 | 664 | 673 | 235 | 1380 | 496 | 1390 | 543 | 1740 | 409 | 1580 | 113 | 305 |
| SP T3-3B | 664 | 296 | 480 | 116 | 1020 | 504 | 1710 | 415 | 1550 | 250 | 1060 | 132 | 456 |
| SP T2-7 | 1140 | 408 | 665 | 235 | 1400 | 431 | 1410 | 425 | 1250 | 242 | 651 | 143 | 455 |
| KY T2-3 | 513 | 343 | 530 | 133 | 1140 | 359 | 1410 | 1.58 | 896 | 119 | 494 | 67.8 | 224 |
| CH T2-1 | 1080 | 537 | 418 | 236 | 1420 | 790 | 2670 | 2.82 | 1570 | 127 | 657 | 46.6 | 174 |
| CH YED3, 0-10 cm | 1420 | 352 | 538 | 281 | 1850 | 722 | 2010 | 651 | 1790 | 642 | 1580 | 730 | 1971 |
| CH Medv grass | 3640 | 855 | 691 | 44.1 | 609 | 63.5 | 156 | 26.0 | 224 | 0.122 | 99.3 | 9.91 | 28.1 |
| CH Y4 grass | 4600 | 887 | 966 | 53.6 | 815 | 66.7 | 261 | 28.6 | 232 | 11.5 | 124 | 8.10 | 30.2 |
| ***Ice complex deposits*** | | | | | | | | | | | | | |
| KU EXP 1-3, 212-216 cm | 1750 | 1600 | 4560 | 1460 | 9460 | 2300 | 8930 | 2020 | 5830 | 1030 | 3660 | 293 | 635 |
| CH YED1, 300-305 cm | 10400 | 4030 | 5800 | 2410 | 17100 | 7270 | 18600 | 6610 | 16600 | 5860 | 14800 | 6810 | 18700 |
| CH YED2, 300-305 cm | 665 | 554 | 892 | 263 | 2070 | 1060 | 3070 | 646 | 2340 | 272 | 1310 | 133 | 532 |
| CH YED3, 520-525 cm | 1400 | 769 | 1030 | 252 | 2040 | 910 | 3120 | 644 | 2440 | 266 | 1160 | 124 | 432 |
| KY EXP1, 0-5 cm | 426 | 304 | 447 | 126 | 1220 | 511 | 1970 | 70.4 | 1390 | 133 | 712 | 60.7 | 233 |
| KY EXP2, 110-115 cm | 722 | 539 | 583 | 153 | 1370 | 606 | 2270 | 457 | 1970 | 181 | 1030 | 86.4 | 333 |
| KY EXP3, 185-200 cm | 446 | 313 | 543 | 158 | 1330 | 562 | 2350 | 401 | 1370 | 154 | 743 | 63.1 | 230 |
| CH DY-3A | 920 | 402 | 895 | 108 | 1070 | 294 | 1180 | 184 | 799 | 70.3 | 331 | 34.4 | 100 |
| CH DY-4A | 327 | 200 | 559 | 74 | 803 | 229 | 1010 | 2.17 | 718 | 64.9 | 334 | 28.7 | 104 |





**Table 4**

Sum of most abundant long-chain $n$-alkanoic acids and $n$-alkanes (concentrations in µg/gOC), and characteristic ratios of $n$-alkanoic acids and $n$-alkanes of topsoil Holocene (modern vegetation/O-horizon) and Pleistocene ice complex samples.

| | $n$-alkanoic acids | | | | $n$-alkanes | | | |
|---|---|---|---|---|---|---|---|---|
| | $\Sigma$HMW[a] ($>C_{22}$) µg/gOC | $\Sigma C_{22}\text{-}C_{28}$ (even) µg/gOC | CPI[b] | HMW acids/ HMW alkanes[a] | $\Sigma$HMW[a] ($>C_{21}$) µg/gOC | $\Sigma C_{25}\text{-}C_{31}$ (odd) µg/gOC | CPI[c] | $C_{25}/(C_{25}+C_{29})$ |
| ***Topsoil (modern vegetation and O-horizon samples)*** | | | | | | | | |
| KU EXP 1-1, 0-16 cm | 3167 | 2670 | 5.8 | 7.1 | 447 | 148 | 2.7 | 0.60 |
| CH YED2, 0-4 cm | 7958 | 6090 | 3.8 | 13 | 612 | 494 | 11 | 0.21 |
| SP T3-3B | 7095 | 5340 | 4.1 | 25 | 280 | 214 | 7.2 | 0.15 |
| SP T2-7 | 6397 | 4700 | 3.7 | 15 | 418 | 323 | 12 | 0.21 |
| KY T2-3 | 4715 | 3940 | 6.8 | 6.6 | 717 | 543 | 9.1 | 0.17 |
| CH T2-1 | 7454 | 6310 | 6.0 | 25 | 300 | 211 | 9.9 | 0.28 |
| CH YED3, 0-10 cm | 11950 | 7230 | 2.9 | 14 | 857 | 647 | 7.8 | 0.09 |
| CH Medv grass | 1216 | 1090 | 9.5 | 5.6 | 217 | 148 | 3.7 | 0.67 |
| CH Y4 grass | 1577 | 1430 | 11 | 7.1 | 223 | 135 | 2.5 | 0.59 |
| *Mean (median)* | *5726 (6397)* | *4310 (4700)* | *5.9* | *13* | *452 (418)* | *318 (214)* | *7.3* | *0.33* |
| *St.dev (IQR)* | *3431 (4290)* | *2190 (4320)* | *2.7* | *7.6* | *230 (332)* | *195 (345)* | *3.6* | *0.22* |
| ***Ice complex deposits*** | | | | | | | | |
| KU EXP 1-3, 212-216 cm | 34854 | 277883 | 4.1 | 39 | 893 | 530 | 4.9 | 0.38 |
| CH YED1, 300-305 cm | 112356 | 67078 | 2.8 | 140 | 806 | 398 | 3.0 | 0.35 |
| CH YED2, 300-305 cm | 11430 | 8791 | 4.1 | 16 | 698 | 405 | 4.6 | 0.33 |
| CH YED3, 520-525 cm | 11145 | 8768 | 4.4 | 14 | 825 | 428 | 3.8 | 0.29 |
| KY EXP1, 0-5 cm | 6293 | 5285 | 6.5 | 10 | 630 | 280 | 2.9 | 0.40 |
| KY EXP2, 110-115 cm | 8293 | 6629 | 4.9 | 12 | 708 | 347 | 2.7 | 0.39 |
| KY EXP3, 185-200 cm | 7196 | 5787 | 4.7 | 11 | 671 | 323 | 3.5 | 0.29 |
| CH DY-3A | 4063 | 3380 | 5.5 | 7.6 | 533 | 344 | 2.7 | 0.35 |
| CH DY-4A | 3295 | 2867 | 8.3 | 9.3 | 355 | 244 | 4.3 | 0.27 |
| *Mean* | *22103 (8290)* | *15160 (6630)* | *5.0* | *29* | *680 (698)* | *367 (347)* | *3.6* | *0.34* |
| *St.dev* | *35150(51140)* | *20880 (3510)* | *1.6* | *43* | *163 (176)* | *85 (81)* | *0.8* | *0.05* |

a HMW: high-molecular weight

b CPI: carbon preference index for chain lengths $C_{22}$-$C_{28}$, for calculation see caption of Fig. 2.

c CPI: carbon preference index for chain lengths $C_{23}$-$C_{31}$, for calculation see caption of Fig. 2.



**Table 5**

δ²H signatures (in ‰) of *n*-alkanoic acids and *n*-alkanes of topsoil Holocene (modern vegetation/O-horizon) and Pleistocene ice complex samples.

| | *n*-alkanoic acids | | | | | | | *n*-alkanes | | | |
|---|---|---|---|---|---|---|---|---|---|---|---|
| | C16 | C18 | C20 | C22 | C24 | C26 | C28 | C25 | C27 | C29 | C31 |
| *Topsoil (modern vegetation and O-horizon samples)* | | | | | | | | | | | |
| KU EXP 1-1, 0-16 cm | -162 | -180 | | -119 | -178 | -203 | -197 | -168 | -240 | -236 | -244 |
| CH YED2, 0-4 cm | -188 | -192 | | -211 | -222 | -232 | -225 | -196 | -237 | -251 | -239 |
| SP T3-3B | | | | -126 | -203 | -218 | -225 | | -125 | -234 | -259 |
| SP T2-7 | -171 | -213 | | -180 | -196 | -210 | -206 | | -182 | -245 | -243 |
| KY T2-3 | | -235 | -185 | -253 | -257 | -264 | -244 | -164 | -240 | -266 | -273 |
| CH T2-1 | -189 | -222 | | -214 | -235 | -236 | -224 | -203 | -221 | -258 | -282 |
| CH YED3, 0-10 cm | -184 | | -190 | -218 | -227 | -225 | -220 | -199 | -234 | -259 | -250 |
| CH Medv grass | -258 | -246 | -253 | -256 | -285 | -286 | | -253 | -236 | -234 | -224 |
| CH Y4 grass | -237 | -244 | -251 | -248 | -245 | -248 | | -227 | -223 | -234 | -209 |
| *Mean* | *-199* | *-219* | *-220* | *-203* | *-228* | *-236* | *-220* | *-201* | *-215* | *-246* | *-247* |
| *St.dev* | *35* | *25* | *37* | *52* | *33* | *27* | *15* | *32* | *39* | *13* | *23* |
| *Ice complex deposits* | | | | | | | | | | | |
| KU EXP 1-3, 212-216 cm | -194 | -227 | -243 | -252 | -245 | -241 | -232 | -237 | -257 | -268 | -265 |
| CH YED1, 300-305 cm | | | -231 | -264 | -271 | -280 | -271 | -217 | -266 | -283 | -297 |
| CH YED2, 300-305 cm | | | | -249 | -262 | -278 | -264 | -254 | -279 | -283 | -307 |
| CH YED3, 520-525 cm | | | -209 | -252 | -266 | -277 | -254 | -243 | -261 | -285 | -305 |
| KY EXP1, 0-5 cm | | | -169 | -260 | -275 | -288 | -273 | -189 | -245 | -269 | -283 |
| KY EXP2, 110-115 cm | -211 | -216 | -252 | -266 | -274 | -294 | -285 | -192 | -254 | -281 | -296 |
| KY EXP3, 185-200 cm | | | -191 | -263 | -277 | -287 | -273 | -210 | -279 | -295 | -309 |
| CH DY-3A | -244 | -256 | -277 | -277 | -277 | -293 | -280 | -195 | -221 | -263 | -298 |
| CH DY-4A | -228 | -229 | -261 | -265 | -262 | -262 | -267 | -251 | -270 | -295 | -313 |
| *Mean* | *-219* | *-232* | *-229* | *-261* | *-268* | *-278* | *-267* | *-221* | *-259* | *-280* | *-297* |
| *Stdev* | *21* | *17* | *37* | *8.6* | *10* | *17* | *16* | *26* | *18* | *12* | *15* |

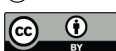



**Table 6**

Location, sampling depth and isotopic values of samples along a surface sediment transect in the Laptev Sea (data from Bröder et al., 2016b), with percentage topsoil (TS) and ice complex deposit (ICD) OC contributions to the samples based on source-apportionment calculations with $\delta^2H$ leaf wax end-members versus $\delta^{13}C$-$\Delta^{14}C$ end-members (end-member values are described in the text).

| | | | | Sample values | | | | | | Source contributions | | | |
|---|---|---|---|---|---|---|---|---|---|---|---|---|---|
| | | | | | | | | | | TS | ICD | TS[c] | ICD[c] |
| ID[a] | Lat | Long | Depth | $C_{27}$ | $C_{29}$ | $C_{31}$ | $C_{27-29-31}$[b] | $\delta^{13}C$ | $\Delta^{14}C$ | using $\delta^2H$ | | using $\delta^{13}C$-$\Delta^{14}C$ | |
| | N | °E | m | ‰ | ‰ | ‰ | ‰ | ‰ | ‰ | | | | |
| TB-46 | 72.700 | 130.180 | 6 | -236.2 | -237.4 | -230.4 | -235.0 | -26.5 | -436 | 89% | 11% | 63% (63%) | 37% (37%) |
| YS-9 | 73.366 | 129.997 | 23 | -233.7 | -231.0 | -227.8 | -231.1 | -26.1 | -415 | 91% | 8.9% | 63% (65%) | 37% (35%) |
| YS-6 | 74.724 | 130.016 | 32 | -234.2 | -241.0 | -235.4 | -236.8 | -25.6 | -465 | 86% | 14% | 51% (59%) | 49% (41%) |
| SW-24 | 75.599 | 129.558 | 46 | -229.3 | -236.5 | -243.5 | -236.4 | -24.8 | -284 | 87% | 13% | 70% (72%) | 30% (28%) |
| SW-23 | 76.171 | 129.333 | 56 | -219.9 | -243.3 | -243.3 | -236.0 | -25.0 | -333 | 83% | 17% | 65% (70%) | 35% (30%) |
| SW-06 | 77.142 | 127.378 | 92 | -219.5 | -237.0 | -241.4 | -233.2 | -23.2 | -364 | 87% | 13% | 39% (53%) | 61% (47%) |
| SW-03 | 78.238 | 126.150 | 2601 | -221.1 | -238.0 | -247.7 | -235.9 | -22.6 | -426 | 85% | 15% | 23% (42%) | 77% (58%) |
| SW-01 | 78.942 | 125.243 | 3146 | -223.8 | -241.8 | -246.0 | -238.0 | -22.3 | -418 | 83% | 17% | 21% (42%) | 79% (58%) |

a Location, depth and bulk carbon isotope data from Bröder et al. (2016b)

b weighted average based on individual concentrations

c numbers in brackets are source contributions using the $\delta^{13}C$-$\Delta^{14}C$ approach but with additional corrections for cross-shelf lateral transport time of topsoil OC (similar as in Bröder et al., 2016a); we applied linear aging along the transect based on the distance from the coast, with a maximum aging of 5000 years for station SW-01.