# Peer review of "Distinguishing between old and modern permafrost sources with compound- specific δ2H analysis"

_The Cryosphere, 2017_

## Referee Comment (RC1) · Anonymous Referee #1 · 14 Mar 2017

This is an extraordinary set of data, where the authors have collected a set of samples in an area not much studied, and analyzed these samples using impressive analytical techniques. The data are well presented and statistically treated and the manuscript is easy to follow. Unfortunately the results are not that exiting as most of the results mainly confirm earlier findings. The few new conclusions are not striking as both the variability within the data are substantial and source values used in the assessment have uncertainties. Nevertheless publication of this contribution could be justified because of the rareness of the data. I leave it to the editor to make the final judgment on this. If a positive decision is made I have no other detailed comments than the references need to be looked over. Some of them even lack publication year.

---

## Referee Comment (RC2) · Anonymous Referee #2 · 28 Apr 2017

I have read the manuscript "Distinguishing between old and modern permafrost sources with compound-specific d2H analysis" by Vonk et al. The study assesses the use of n-alkanes and n-alkanoic acid distribution patterns and $\delta$2H values to inform on the sources of sedimentary organic matter derived from different types of permafrost melt. They apply this method in combination with bulk organic geochemistry, and compare this approach with a source identification method based on $\delta$13C and radiocarbon from bulk sediments. The authors identify strengths and weaknesses of each approach, and conclude that each approach has its own merits and drawbacks. Overall I found the manuscript to be adequately written, the experimental design and execution to be sound, and the analysis and interpretation to be supported by the data.

[Figure]

I have a few comments that I've outlined below where I think that some clarification would help, and a few stylistic suggestions, but otherwise I have no major critiques. The topic is relevant and within the scope of the journal and I recommend publication following consideration of mu comments.

General comments:

I think the abstract is too long, and that it goes into too much specific detail. I think that it could and should be made more succinct.

The phrase "molecular-bulk upscaling challenge" is used without enough introduction/definition. I understand what you mean by it, but I think that it would be better to explain what this is exactly in a bit more detail.

A general comment about the structure of the discussion is the separation of the 13C-14C data from the bulk geochemistry. Why are these measurements not included in this grouping? If you measure 13C or 14C on a bulk sample, isn't that "bulk geochemistry"? You might be able to circumvent this issue just by renaming the bulk section to "bulk elemental geochemistry" or something like that.

One thing that I think is also missing from the discussion is some mention of the possibility that the terrestrial sampling density may have missed some of the possible heterogeneity in permafrost chemistry. I realize that it's not easy to sample in this part of the world, but is there any reason to think that the results might look different if you had soil samples from 50 more sites? Why or why not? This would apply to the $\delta$2H data, as well as the other data.

Specific comments:

Line 78 – change "into" to "in"

Line 99 – Personally, I'm not a fan of non-standard acronyms like this (ICD in this case). They require an elevated level of buy in from the reader, which I think takes away from the accessibility of the manuscript. That's just my opinion, there's plenty of precedent

for this kind of thing of course.

Line 137 – At the introduction of the $\delta$2H discussion, it might help to frame the study better if you begin by saying that you propose the new tool, as well as evaluate the performance using a suite of other geochemical data including the aforementioned $\delta$13C-radiocarbon method.

Line 142-143 - These are nice papers, but they aren't really the best references to support the assertion that "the isotopic value of local precipitation is a function of local climate"

Line 149 - If you mean to give the maximum range you could point out that precip in east africa can be upwards of +50 per mil, while the SLAP2 (Standard Light Antarctic Precipitation 2) standard is -427.5 per mil.

Line 162-168 – The end of the introduction falls a little flat in my opinion. At the moment you say what you do in your study, followed by a general statement about why it's important to study these types of questions. What's missing to me is a statement that directly comments on how what you do with this study will help with these important questions. As it is currently written it doesn't setup the next section so effectively.

Line 195 - I think it is better to replace your internal lab sample codes with something more straightforward when reporting the results (things like "CH DY-3A" are meaningless to the reader and hard to remember). Include them in a data file or something if you want to be able to cross-reference with Vonk et al., 2013, but for presentation purposes I would simplify.

Line 232 – Remember to define acronyms at first use.

Line 243 – Check super/subscripting for H3+.

Line 244 – Give units for H3+.

Line 248 - The "methylation effect" language is odd to me, since it makes it sound like

what was quantified was the difference in $\delta$2H values between the derivatized and non derivatized standard, rather than the $\delta$2H value of the hydrogen in the added methyl group. Since the magnitude of the "methylation effect" will be different depending on what the $\delta$2H value of the covalently bonded hydrogen in the methylated fatty acid is in addition to the chain length, you want to do the correction by mass balance. Probably that is what you did, but the language doesn't make it sound that way.

Line 251 – This call to table 5 is out of order since you haven't called tables 2-4 yet.

Line 259 – Not sure what you mean exactly by "with mean and standard deviations obtained from the literature values".

Line 296 – I might add a few words to the start of the sentence that begins on this line to make it clear that you are discussing distributions within individual samples, and that you are still talking about topsoil samples only.

Line 318 – This call to table 3 is out of order.

Line 354 – spell check.

Line 417 – change "proxies" to "proxy"

Line 452 – This is the first mention of results from the shelf-slope samples. In the methods you point out a reference for more information on the sampling procedures, but what about the laboratory analyses and results? This should be included in the earlier sections.

Line 454 - I like how you use the individual n-alkanes rather than arbitrarily averaging them together.

Lines 481 – 500 – Somewhere in this section, or somewhere else in the manuscript if it fits better, it would be good to discuss how variability within an end member might impact the results. This is important for both the $\delta$2H and the $\delta$13C-radiocarbon approaches, but it seems like it would be especially important for the radiocarbon. In

addition to the acknowledged aging along the transect won't there be different ages within a topsoil permafrost? How might this impact the results if melting/erosion occurs at different depths/ages within a site?

Line 568 – As with the end of the introduction, I think that the end of the conclusion could go a little further to bring this study back together with the big picture goals. Remind us how "increasing our understanding of the fate of thawing permafrost in the coastal environment" will help us and why we should care about it.

Figure 2 – I would add the color legend to this figure that you already use on the other figures. I would also list n values in the caption or on the figure.

Figure 5 - I would list the modern/ICD labels as headers rather than within the data.

---

## Referee Comment (RC3) · T.J. Porter (Referee) · 28 Apr 2017

Vonk et al. evaluate the potential to use stable hydrogen isotope ratios of fossil terrestrial plant n-alkyls as a tracer for shallow vs. deep carbon sources, which are later blended and deposited on the Siberian continental shelf. This work is important for a number of reasons, including understanding the impact of interglacial warm climates on the stability of the cryosphere and northern landscapes, and with obvious implications for climate-carbon feedbacks. The authors compare this novel compound-specific proxy to other carbon source proxies already known in the literature, and they do a very good job of highlighting the advantages/disadvantages of each. While the sample size was small, the analysis was rigorous and accounted for the major uncertainties. The

authors also discussed the influence of various uncertainties and assumptions on their end-member modeling results. They follow this with thoughtful discussion on how to advance this research future. The manuscript + figures + tables are well formatted and easy to follow. I believe this work is well suited for publication in the Cryosphere. I have only a few general and specific comments that I would like the authors to address, and following these very minor revisions this paper should be accepted.

General comments: -Title should indicate the study region (e.g., Laptev Sea catchment, NE Siberia).

-Abstract is one of the longest I've read in recent years. It includes a lot of useful information, but could (and probably should) be more concisely written. I leave this to authors and editor to decide.

-There is now a large body of literature on the isotopic composition of relict ice from ICD's in this very same region (see Opel et al. 2017 Climate of the Past; and references therein). These studies find that precipitation isotope composition recorded in these ICD's was highly variable during Pleistocene cold stages; for example, texture and pore ice $\delta2H$ values range from roughly -250‰ to -160‰ between ca. 50-30 cal ka BP, while ice wedge values (winter precip) during the same interval range from roughly -260 to -230‰İf the fossil plants were using the same water that is preserved in the pore ice, then there may be a significant amount of variance not yet captured in the n-alkyl dataset from the (n = 9) ICDs sampled in this study. The spatial distributions of distinct ICD units in this region are not equal (see Opel et al., 2017) and, thus, have different potentials for erosion and contribution to the blend of n-alkyls deposited on the shelf. I would like the authors to acknowledge this potentially major source of uncertainty. I would also ask the authors disclose any information they have on the age of the sampled ICDs and, if possible, cross-reference to the regional stratigraphy scheme outlined in Opel et al. (2017).

-This paper would benefit from another figure that provides photographic examples of

the ICDs and topsoil sections.

Specific comments: L44, The n-alkane sum and interquartile range given (210±350 ug/gOC) implies negative concentrations are possible, and is not consistent with Figure 2a. This also occurs on L299.

L149, instead of citing the IAEA website, better to cite a peer-reviewed article that supports your statement. Dansgaard (1964, Tellus) is appropriate.

L158, it might be worth stating the underlying assumption, that colder air temperatures during the Pleistocene generally correlate with 2H-depleted precipitation; therefore, long-chain n-alky $\delta$2H during Pleistocene cold-stages should also be depleted compared to present. Also note that 'colder' and 'drier' could have opposing effects. All other factors equal (e.g., biochemical fractionation), a drier atmosphere during Pleistocene cold-stages could result in a larger leaf water enrichment and $\delta$2H n-alkyls (if RH is lower, despite lower air temps), which would lessen the overall offset between modern and Pleistocene n-alkyl $\delta$2H.

L199-201, if species information is available for the grasses and birch, please indicate.

L308, the sphagnum index could also include C23 (see Bush and McInerney, 2013, GCA). For modern sphagnum samples I've collected in NW Canada (>65°N), C23 is usually abundant (unpublished data). This suggestion isn't critical, but might be a more accurate metric for sphagnum vs. woody plants.

L345, please delete 'it seems'. If there is uncertainty, this can be described in a more quantitative way.

L519-521, unclear if you are talking about potential overprinting of the fossil $\delta$2H in situ (e.g., with water in the frozen ICD), or following transport and deposition on the shelf. Please clarify. Also, give a citation that supports the statement that overprinting is enhanced in low pH environments.

Trevor Porter.

---

## Author Comment (AC1) · 3 Jun 2017

**Response to review comments of manuscript " Distinguishing between old and modern permafrost sources with compound-specific $\delta^2$H analysis"**
The Cryosphere Discussions doi:10.5194/tc-2017-17

We thank the three reviewers for their comments, and provide answers to these comments and suggestions below.

Black: review comment
*Blue, italic: our response and edits in manuscript*
* * *
**Reviewer Trevor Porter**
General comments:
Title should indicate the study region (e.g., Laptev Sea catchment, NE Siberia).
*We have added "in the Northeast Siberian land-shelf system".*

Abstract is one of the longest I've read in recent years. It includes a lot of useful information, but could (and probably should) be more concisely written. I leave this to authors and editor to decide.
*We have shortened the abstract, as indeed it was rather long.*

There is now a large body of literature on the isotopic composition of relict ice from ICD's in this very same region (see Opel et al. 2017 Climate of the Past; and references therein). These studies find that precipitation isotope composition recorded in these ICD's was highly variable during Pleistocene cold stages; for example, texture and pore ice 2H values range from roughly -250‰ to -160‰ between ca. 50-30 cal ka BP, while ice wedge values (winter precip) during the same interval range from roughly -260 to -230‰. If the fossil plants were using the same water that is preserved in the pore ice, then there may be a significant amount of variance not yet captured in the n-alkyl dataset from the (n = 9) ICDs sampled in this study. The spatial distributions of distinct ICD units in this region are not equal (see Opel et al., 2017) and, thus, have different potentials for erosion and contribution to the blend of n-alkyls deposited on the shelf. I would like the authors to acknowledge this potentially major source of uncertainty. I would also ask the authors disclose any information they have on the age of the sampled ICDs and, if possible, cross-reference to the regional stratigraphy scheme outlined in Opel et al. (2017).
*Thank you for highlighting this point. We had not seen the article by Opel et al. While it is still not a peer-reviewed published paper (status listed as "in discussion"), it clearly provides additional insight to ICD from this region. We have now cited this paper, as well as a few others in the Introduction (and elsewhere) of our revised paper. We agree that the point brought up can be a source of uncertainty, and have acknowledged this fact in section 4.2 quite extensively.*
*We have also added the 14C information we have available on a few of the samples analyzed (Table 1). The new text reads as follows (line 492-510):*
*"Finally, we realize that the amount of soil and ICD samples analyzed in this study is limited, and want to point out that the results may change when more data are analyzed in the near future. Additionally, studies have shown that the $d^2$H signature of ice within ICD permafrost deposits can range from roughly -150‰ to -260‰ depending on the type of ice (wedge ice vs. texture ice) as well as the period of formation (different Pleistocene cold stages) (Opel et al., 2017 and references therein). The source of water (i.e. type of ice) and age of the deposit will therefore influence the n-alkane or n-alkanoic acid $d^2$H signal. However, regardless of the natural variability associated with the processes mentioned above, both ICD and texture-ice isotopic compositions appear to reflect long-term climate changes (Opel et al., 2017; Schwamborn et al., 2006; Dereviagin et al., 2013; Porter et al., 2016) which, likely, were also captured in the n-alkane or n-alkanoic acid d2H signal. Unfortunately, we do not have $^{14}$C-ages available for all ICD samples, so cross-referencing to published stratigraphies in the region is not possible. Coastal sediments, however, will represent a*

*mixture of material released from different depths, outcrops, and stratigraphies within the catchment or coast. For source-apportionment applications, we reason that a growing body of leaf wax d$^2$H end-member data from the ICD region can overcome the variability issues highlighted above."*

This paper would benefit from another figure that provides photographic examples of the ICDs and topsoil sections.
*We do not have high-quality photographic material available from all the sites, unfortunately, but have added a figure as supplementary information with some examples.*

Specific comments:
L44, The n-alkane sum and interquartile range given (210±350 ug/gOC) implies negative concentrations are possible, and is not consistent with Figure 2a. This also occurs on L299.
*Since we shortened the abstract, we have removed the n-alkane concentrations from the abstract (L44) but have changed the notation in (the previous) L299 as this was perhaps confusing. The interquartile range was 350 (Q1-Q3), so we have chosen to now report this as $210_{148}^{494}$ to make it more clear that IQR1 is 148 and IQR3 is 494. (lines 344-349 and lines 364-369). We have also adjusted Table 4.*

L149, instead of citing the IAEA website, better to cite a peer-reviewed article that supports your statement. Dansgaard (1964, Tellus) is appropriate.
*OK, good suggestion, we have done this (line 160).*

L158, it might be worth stating the underlying assumption, that colder air temperatures during the Pleistocene generally correlate with 2H-depleted precipitation; therefore, long-chain n-alky 2H during Pleistocene cold-stages should also be depleted compared to present. Also note that 'colder' and 'drier' could have opposing effects. All other factors equal (e.g., biochemical fractionation), a drier atmosphere during Pleistocene cold-stages could result in a larger leaf water enrichment and 2H n-alkyls (if RH is lower, despite lower air temps), which would lessen the overall offset between modern and Pleistocene n-alkyl 2H.
*Thanks, this is a good suggestion. We have edited this sentence to now read "Despite the plant fractionation associated with kinetic and plant physiology (Sachse et al., 2012), we hypothesize that $\delta^2$H signatures of leaf wax n-alkanoic acids and n-alkanes are more depleted in OC from permafrost deposits formed during the colder Pleistocene (generally correlating with $^2$H-depleted precipitation), compared to more enriched values in OC from active layer or surface permafrost formed during the warmer Holocene. " (lines 178-183)*

L199-201, if species information is available for the grasses and birch, please indicate.
*Both samples were grass samples, but one of them was collected in the tundra, and one of them further south in a birch forest. We have clarified this (line 227-229). Unfortunately, we do not have species information available.*

L308, the sphagnum index could also include C23 (see Bush and McInerney, 2013, GCA). For modern sphagnum samples I've collected in NW Canada (>65N), C23 is usually abundant (unpublished data). This suggestion isn't critical, but might be a more accurate metric for sphagnum vs. woody plants.
*We are aware that either C23 or C25 can be abundant, but meant to illustrate the general differences (very small) between the two sample types. But, for comparison we have now replaced the C25/(C25+C29) ratio with C23/(C23+C29) in Figure 2 and added the C23 ratio to Table 4. Also, we have added a sentence to the text (line 357-360). Here, the average values for topsoil vs. ICD samples are further apart, yet still not statistically significant.*

L345, please delete 'it seems'. If there is uncertainty, this can be described in a more quantitative way.
*We have deleted it (line 407).*

L519-521, unclear if you are talking about potential overprinting of the fossil 2H in situ (e.g., with water in the frozen ICD), or following transport and deposition on the shelf. Please clarify. Also, give a citation that supports the statement that overprinting is enhanced in low pH environments.
*We have clarified this by stating that the environmental water can be coming from various sources (e.g. in situ or during transport after thaw). Also, we have removed the statement on low pH environments as we could not support this with a proper reference, and it is less relevant to our study (lines 603-607).*
* * *
**Anonymous reviewer #1**
If a positive decision is made I have no other detailed comments than the references need to be looked over. Some of them even lack publication year.
*We have carefully read through the references list and edited/corrected where needed.*
* * *
**Anonymous reviewer #2**
General comments:
I think the abstract is too long, and that it goes into too much specific detail. I think that it could and should be made more succinct.
*We agree. This point was also brought up by Reviewer Trevor Porter. We have shortened the abstract.*

The phrase "molecular-bulk upscaling challenge" is used without enough introduction/ definition. I understand what you mean by it, but I think that it would be better to explain what this is exactly in a bit more detail.
*We have added a bit more specific description the first time we use this definition, in the introduction: " This $\delta^{13}C$-$\Delta^{14}C$ dual-carbon isotope approach carries the strong advantage that it operates on the bulk OC level, thereby circumventing the "molecular-bulk upscaling challenge". This challenge relates to issues associated with upscaling from the molecular isotope level to the bulk level. These issues relate to the relative concentration (n-alkanes and n-alkanoic acids represent only a fraction of the total OC) but also to processes such as selective degradation, differences in physical association, or dispersion differences. " (lines 132-138).*

A general comment about the structure of the discussion is the separation of the 13C-14C data from the bulk geochemistry. Why are these measurements not included in this grouping? If you measure 13C or 14C on a bulk sample, isn't that "bulk geochemistry"?
You might be able to circumvent this issue just by renaming the bulk section to "bulk elemental geochemistry" or something like that.
*We do not really follow the reviewer here. In the first section of the discussion we talk about %C, C/N values, as well as d13C on bulk samples. As such, we named this section " ... bulk geochemistry ... as it includes both elemental and isotopic measurements on bulk samples. The same is the case for the first section of the Results.*

One thing that I think is also missing from the discussion is some mention of the possibility that the terrestrial sampling density may have missed some of the possible heterogeneity in permafrost chemistry. I realize that it's not easy to sample in this part of the world, but is there any reason to think that the results might look different if you had soil samples from 50 more sites? Why or why not? This would apply to the 2H data, as well as the other data.

*This is a valid point. We have added the following text to the manuscript, at the end of section 4.2: "Finally, we realize that the amount of soil and ICD samples analyzed in this study is limited, and want to point out that the results may change when more data are analyzed in the near future." (line 492-494)*
*Regarding the second point/suggestion of this review comment, we think it would be too speculative to give more detail regarding the possible differences in results if more data were to be obtained.*

Specific comments:
Line 78 – change "into" to "in"
*Changed.*

Line 99 – Personally, I'm not a fan of non-standard acronyms like this (ICD in this case). They require an elevated level of buy in from the reader, which I think takes away from the accessibility of the manuscript. That's just my opinion, there's plenty of precedent for this kind of thing of course.
*We realize there are different opinions with respect to acronym usage, but prefer to continue using it as this shorter version is commonly used and it also improves readability.*

Line 137 – At the introduction of the 2H discussion, it might help to frame the study better if you begin by saying that you propose the new tool, as well as evaluate the performance using a suite of other geochemical data including the aforementioned 13C-radiocarbon method.
*This is a good suggestion. We have now in the revised ms better introduced our tools in the introduction: " We will evaluate the performance of this complementary tool using additional geochemical data as well as the bulk $\delta^{13}C$-$\Delta^{14}C$ mixing approach." (lines 148-150).*

Line 142-143 - These are nice papers, but they aren't really the best references to support the assertion that "the isotopic value of local precipitation is a function of local climate"
*Yes, we agree. We have added: Craig H. 1961. Isotopic variations in meteoric waters. Science 133: 1702–1703. (line 154)*

Line 149 - If you mean to give the maximum range you could point out that precip in east africa can be upwards of +50 per mil, while the SLAP2 (Standard Light Antarctic Precipitation 2) standard is -427.5 per mil.
*We have edited this to now present the maximum range, using the values/locations that this reviewer provides (lines 160-163).*

Line 162-168 – The end of the introduction falls a little flat in my opinion. At the moment you say what you do in your study, followed by a general statement about why it's important to study these types of questions. What's missing to me is a statement that directly comments on how what you do with this study will help with these important questions. As it is currently written it doesn't setup the next section so effectively.
*We see the point, and have therefore now added one more sentence that specifically mentions the use of our proposed tool, at the very end of the introduction: "Our proposed tool may be used to trace these temporal and spatial differences in OC release from permafrost thaw, as well as the extent of burial of OC in sedimentary reservoirs."(lines 191-193).*

Line 195 - I think it is better to replace your internal lab sample codes with something more straightforward when reporting the results (things like "CH DY-3A" are meaningless to the reader and hard to remember). Include them in a data file or something if you want to be able to cross-reference with Vonk et al., 2013, but for presentation purposes I would simplify.
*We have renamed the samples with TS-1, TS-2, ICD-1, ICD-2 etc, and include the original sampling ID in Table 1.*

Line 232 – Remember to define acronyms at first use.
*This acronym was defined in the previous paragraph.*

Line 243 – Check super/subscripting for H3+.
*We have edited this into $H_3^+$ (line 276).*

Line 244 – Give units for H3+.
*We have added units (‰ per V) and have also added a reference for the use of $H_3^+$ (Sessions et al., 2011) (line 277).*

Line 248 - The "methylation effect" language is odd to me, since it makes it sound like what was quantified was the difference in 2H values between the derivatized and non derivatized standard, rather than the 2H value of the hydrogen in the added methyl group. Since the magnitude of the "methylation effect" will be different depending on what the 2H value of the covalently bonded hydrogen in the methylated fatty acid is in addition to the chain length, you want to do the correction by mass balance. Probably that is what you did, but the language doesn't make it sound that way.
*We agree, we have changed this into "methylation correction" (lines 281-282).*

Line 251 – This call to table 5 is out of order since you haven't called tables 2-4 yet.
*This sentence is a general remark on how we report the d2H values, so we decided to not call table 5 at this particular place.*

Line 259 – Not sure what you mean exactly by "with mean and standard deviations obtained from the literature values".
*This was meant to refer to the end-member values for the d13C-D14C source apportionment, but is perhaps confusing. We have now specified this to "with mean and standard deviations obtained from our analysis (d2H on TS and ICD samples) and from literature (13C and 14C on end-members)" (lines 298-299).*

Line 296 – I might add a few words to the start of the sentence that begins on this line to make it clear that you are discussing distributions within individual samples, and that you are still talking about topsoil samples only.
*We have done this by adding " for Topsoil-PF samples" to this sentence (line 343).*

Line 318 – This call to table 3 is out of order.
*Indeed, this should be table 2. In the previous sentence, however, we have changed a call to table 2 into table 3. As all tables have already been called before, we did not change the order of the tables.*

Line 354 – spell check.
*Thanks, we have corrected this.*

Line 417 – change "proxies" to "proxy"
*We have changed this.*

Line 452 – This is the first mention of results from the shelf-slope samples. In the methods you point out a reference for more information on the sampling procedures, but what about the laboratory analyses and results? This should be included in the earlier sections.
*Yes, good point. We have added a brief paragraph at the end of section 2.2 (lines 286-290).*

Line 454 - I like how you use the individual n-alkanes rather than arbitrarily averaging them together.
*Thanks.*

Lines 481 – 500 – Somewhere in this section, or somewhere else in the manuscript if it fits better, it would be good to discuss how variability within an end member might impact the results. This is important for both the 2H and the 13C-radiocarbon approaches, but it seems like it would be especially important for the radiocarbon. In addition to the acknowledged aging along the transect won't there be different ages within a topsoil permafrost? How might this impact the results if melting/erosion occurs at different depths/ages within a site?
*We agree that variability within the end-members plays an important role and should be taken into account. For the 13C and 14C approaches, the amount of end-member data available is fairly good (and growing) with 30-40 data points for 13C and >300 data points for 14C. This is described in the second paragraph of section 4.3. Our Markov Chain Monte Carlo mass-balance model actually accounts for the end-member variability (described and referenced in section 3.2). When thawing and erosion occurs at different depths or ages, at various locations throughout a watershed or coastline, the signal will be averaged when measured in coastal sediments. We have now acknowledged this important point (lines 505-507) (reviewer Trevor Porter also posted a related comment). Regarding the amount of d2H end-member data available: we are aware that our sample set only exists of n=9 data points for each source. Variability in the mean end-member values may therefore change (or, perhaps, decrease) when more data become available. We have now also briefly mentioned this point at the end of section 4.2 (lines 492-494).*

Line 568 – As with the end of the introduction, I think that the end of the conclusion could go a little further to bring this study back together with the big picture goals. Remind us how "increasing our understanding of the fate of thawing permafrost in the coastal environment" will help us and why we should care about it.
*We have added a bit more "big picture" text to place the results of our study into context with the general goals outlined in the introduction. The final paragraph of the conclusions now reads:*
*"This study shows that $\delta^2H$ of leaf wax molecules has the potential to be used in quantitative source-apportionment studies of thawing permafrost in coastal or marine settings. It can serve as an alternative or complementary approach to the commonly applied bulk $\delta^{13}C$-$\Delta^{14}C$ method. We recommend continuing data collection and optimization of end-member definition and calibration. Refining the molecular $d^2H$ proxy presented here will be beneficial in pinpointing the location and extent of OC release from thawing permafrost in the coastal or fluvial environment. With enhanced Arctic warming and associated intensification of permafrost thaw, constraining the amount and fate of permafrost OC release will help to assess the magnitude of the permafrost carbon feedback to climate warming." (lines 658-663).*

Figure 2 – I would add the color legend to this figure that you already use on the other figures. I would also list n values in the caption or on the figure.
*Yes, this has been changed.*

Figure 5 - I would list the modern/ICD labels as headers rather than within the data.
*OK, we have done this.*

---

## Author Comment (AC2) · 3 Jun 2017

The comment was uploaded in the form of a supplement:
http://www.the-cryosphere-discuss.net/tc-2017-17/tc-2017-17-AC2-supplement.pdf

---

## Author Comment (AC3) · 3 Jun 2017

The comment was uploaded in the form of a supplement:
http://www.the-cryosphere-discuss.net/tc-2017-17/tc-2017-17-AC3-supplement.pdf

---

## Author Comment (AC4) · 3 Jun 2017

**Distinguishing between old and modern permafrost sources in the Northeast Siberian land-shelf system with compound-specific $\delta^2$H 
[revised manuscript text omitted]

challenge". This challenge relates to issues associated with upscaling from the
molecular isotope level to the bulk level. These issues relate to the relative
concentration (*n*-alkanes and *n*-alkanoic acids represent only a fraction of the total
OC) but also to processes such as selective degradation, differences in physical
association, or dispersion differences. However, the $\delta^{13}$C-$\Delta^{14}$C approach also has
drawbacks, such as a weak distinction between the $\delta^{13}$C end-member values of
Topsoil-PF versus ICD-PF. Also, the marine $\delta^{13}$C end member values in coastal Arctic
shelf waters are uncertain and may be more depleted than at mid-latitudes due to
uptake of relatively depleted dissolved $CO_2$ values caused by cold polar water
(Meyers, 1997; Tesi et al. *this special issue*) or degradation of terrestrial matter
(Anderson et al., 2009; 2011; Semiletov et al., 2013; 2016), generating a potential
overlap between marine and topsoil $\delta^{13}$C end-members.
Here we propose a complementary tool to trace permafrost OC release into the
coastal environment based on molecular $\delta^{2}$H analysis on leaf waxes. We will evaluate
the performance of this tool using additional geochemical data as well as the bulk
$\delta^{13}$C-$\Delta^{14}$C mixing approach. Isotopes in water molecules ($\delta^{2}$H or $\delta^{18}$O) in glacial ice
cores as well as in massive ground ice in the northern hemisphere have been used for
reconstructing palaeotemperatures (e.g., Kotler and Burn, 2000; Johnson et al., 2001;
Opel et al., 2011; Meyer et al., 2015; Wetterich et al., 2016) as the isotopic value of
local precipitation is a function of local climate (Craig, 1961; Sachse et al., 2004; Smith
and Freeman, 2006). Higher plants use water as their primary source of hydrogen
during photosynthesis (Sternberg, 1988). The $\delta^{2}$H isotope values of leaf wax *n*-
alkanoic acids or *n*-alkanes are therefore reflecting the $\delta^{2}$H isotopic value of local
precipitation (e.g., Sachse et al., 2004; Sessions et al., 1999), after correction for the
net fractionation during biosynthesis, and evapotranspiration (Leaney et al., 1985).
Global precipitation values can vary immensely (Dansgaard, 1964) with values up to
+50‰ in Eastern Africa but approaching -200‰ near the North Pole (www.iaea.org)
or even below -400‰ in Antarctica (i.e. SLAP2 standard, Standard Light Antarctic
Precipitation, is -427.5‰). Additionally, the fractionation between source water and
plant wax molecules varies both in time and space, and can be up to -170‰ (Smith
and Freeman, 2006; Sachse et al., 2004; Polissar and Freeman, 2010) but appears
relatively small at higher latitudes (between -59 and -96‰; Shanahan et al., 2013;
Wilkie et al., 2013; Porter et al., 2016). Differences in $\delta^{2}$H signatures of leaf wax
molecules from terrestrial regions with different (past) climates could therefore
potentially be applied to derive the relative proportion of different types of thawing permafrost in nearby coastal settings. Despite the plant fractionation associated with kinetics and plant physiology (Sachse et al., 2012), we hypothesize that $\delta^2$H signatures of leaf wax $n$-alkanoic acids and $n$-alkanes are more depleted in OC from permafrost deposits formed during the colder Pleistocene (generally correlating with $^2$H-depleted precipitation), compared to more enriched values in OC from active layer or surface permafrost formed during the warmer Holocene.

This study investigates a source-specific $\delta^2$H signature for both ICD permafrost and recent, surface soil permafrost in Northeast Siberia. Furthermore, we explore the possibilities of using these isotopic end-member values in regional source-apportionment calculations that aim to quantify the relative contribution of different sources of permafrost OC. As permafrost thaw progresses, particularly in ice-rich permafrost such as ICD, it is increasingly important to trace the fate of remobilized and decomposing OC in the Arctic environment. Our proposed tool may be used to trace these temporal and spatial differences in OC release from permafrost thaw, as well as the extent of burial of OC in sedimentary reservoirs.

**2    Methods**

**2.1    Sampling**

A total of 18 samples were collected throughout the Siberian Arctic. Recent surface soils (n=7) and vegetation (n=2) samples were analyzed and (from here on) referred to as the "topsoil" permafrost (Topsoil-PF) sample set, whereas ICD-PF samples were obtained from ICD soil profiles (n=7) and suspended particulates from ICD formations (n=2) (Fig. 1 and Table 1). Eight offshore sediments along a shelf-slope-continental rise transect in the Laptev Sea were collected in 2014, further marine sampling details can be found in Bröder et al. (2016b).

The Topsoil-PF samples represent O and A soil genetic horizons in sites with active soil formation. The sites where chosen to represent typical soil and vegetation types in the investigated permafrost landscapes, including both taiga and tundra sites. Samples were collected by depth or soil horizon increments from open soil pits using fixed volume sampling procedures.

The ICD-PF samples were collected from vertical exposures that were excavated to expose intact permafrost. Fixed-volume samples were collected by coring horizontally into the frozen sediments to extract ICD-PF samples from consecutive depths.

For more details about sampling sites, including location, vegetation and soil types see table 1 (terminology following the U.S.D.A. Soil Taxonomy; Soil Survey Staff, 2014). Sampling was done in late summer near the time of maximum annual active layer depth, in July 2010 (ICD-8 and ICD-9; Vonk et al. (2013)) and August 2011 (Palmtag et al., 2015) for the Kolyma River region, in August 2012 for the lower Lena River and Indigirka River (Siewert et al., 2015; Weiss et al., 2015) and in August 2013

for the upper Lena River (Siewert et al., 2016).  For more detailed descriptions of
sample collection we refer to these references.  The vegetation samples TS-8G (grass)
and TS-9G (grass) were obtained from the tundra near Medvezhka River and a birch
forest near Y4 stream, respectively, in July 2012.
Samples JCD-8 and JCD-9 were obtained in July 2010 at the Duvannyi Yar ICD
exposure along the Kolyma River (Vonk et al., 2013). The particulate sediment
samples were taken from thaw streams that were freshly formed from thawing ICD
(transport time from thaw to sampling estimated to be less than 1h).

**2.2    Analytical methods**

Freeze-dried samples were extracted using an ASE 200 accelerated solvent extractor
(Dionex Corporation, USA) using DCM/MeOH (9:1 v/v) at 80°C ($5x10^6$ Pa)
(Wiesenberg et al., 2004). After the extraction, solvent-rinsed activated copper and
anhydrous sodium sulfate were added to the extracts to remove sulfur and excess
water, respectively. After 24 h, extracts were filtered on pre-combusted glass wool
and concentrated with the rotary evaporator. Extracts were transferred into glass
tubes, evaporated to complete dryness and re-dissolved in 500 µl of DCM. Lipid
fractionation was performed via column chromatography using amino-propyl Bond
Elut (500 mg/3 ml) to retain the acid fraction and $Al_2O_3$ to separate the hydrocarbon
and polar fractions (Vonk et al., 2010).
Prior to the analyses, saturated *n*-alkanes (hydrocarbon fraction) were further
purified using 10% $AgNO_3$ coated silica gel to retain the unsaturated fraction. The acid
fraction was methylated using a mixture of HCl, MilliQ water and methanol at 80°C
overnight to obtain the fatty acid methyl ester (FAME) fraction. Methylated acids
were extracted with hexane and further purified using 10% $AgNO_3$ coated silica gel.
The hydrocarbon and FAME fractions were quantified via gas chromatography mass
spectrometry (GC–MS) in full scan mode (50-650 m/z) using the response factors of
commercially available standards (Sigma-Aldrich). The GC was equipped with a 30
m×250 µm DB5-ms (0.25 µm thick film) capillary GC column. Initial GC oven
temperature was set at 60°C followed by a 10°C $min^{-1}$ ramp until a final temperature
of 310°C (hold time 10 min).
The hydrogen-isotopic composition of hydrocarbon and FAME fractions was
measured with continuous-flow GC - isotope ratio - MS. Purified extracts were
concentrated and injected (1-2 µl) into a Thermo Trace Ultra GC equipped with a
30m×250 µm HP5 (0.25 µm thick film) capillary GC column. Oven conditions were
similar to the setting used for the quantification. The conversion of organic
biomarkers to elemental hydrogen was accomplished by high-temperature
conversion (HTC) at 1420°C (Thermo GC Isolink). After the HTC, $H_2$ was introduced
into the isotope ratio MS (Thermo Scientific™ Delta V™IRMS) for compound-specific
determination of $\delta^2H$ values via a Thermo Conflo IV. Following a linearity test, we only
used peaks with amplitude (mass 2) between 1500 and 8000 mV for the evaluation.
The $\delta^2H$ values were calibrated against saturated HMW *n*-alkanes using the reference substance mix A4 (Biogeochemical Laboratories, Indiana University). The $H_3^+$ factor
(Sessions et al., 2001) was determined every day and stayed constant (<3 ‰/V )
throughout our analyses period. Each purified extract was injected three times.
FAMEs were further corrected to account for the methylation agent by comparing the
hydrogen abundance of lauric acid ($C_{12}$-FA; i.e. 12 carbon atoms) as acid and
corresponding methyl ester. The average methylation correction for lauric acid was
23.97±3.9‰ (n=4). This correction was, normalized to chain length (i.e. increasing
chain lengths result in lower corrections), applied to all the FAMEs. $\delta^2H$ values of $n$-
alkanes and FAMEs are reported as mean, standard deviation and weighted average.
Details of the analytical methods for extraction, work-up, and purification of the eight
offshore sediment samples for biomarker analysis that are included in our source-
apportionment comparison (section 4.3) can be found in Bröder et al. (2016b). The
$\delta^2H$ analysis on the shelf sediments was performed in parallel with the ICD-PF and
Topsoil-PF samples, according to the method described above.
**2.3 Source apportionment**
The compound-specific $\delta^2H$ signatures in this study were used to differentiate
between the two major sources (end-members), Topsoil-PF and ICD-PF, using an
isotopic mass-balance model. We used a Markov chain Monte Carlo (MCMC) approach
to account for the end-member variability (Andersson et al., 2015; Bosch et al., 2015).
The end-members were represented by normal distributions, with mean and
standard deviations obtained from our analysis ($\delta^2H$ on TS and ICD samples) and
from literature ($\delta^{13}C$ and $\Delta^{14}C$ on end-members)". For each Laptev Sea station, the
isotope signatures from three different terrestrial molecular markers (long-chain $n$-
alkanes $C_{27}$, $C_{29}$ and $C_{31}$) were used jointly to improve source apportionment
precision. The $\delta^2H$ signatures for the two end-members were based on our Topsoil-
PF and ICD-PF samples.
The compound-specific $\delta^2H$-based source apportionment was compared to
$\Delta^{14}C/\delta^{13}C$-based analysis of bulk OC using analogous MCMC techniques (e.g., Vonk et
al., 2012). The $\Delta^{14}C/\delta^{13}C$-approach allows estimation of the relative contribution of a
third source, marine, which does not affect the presently investigated (terrestrial)
compounds. Accounting for the marine component to OC allows direct comparison of
the Holocene and Pleistocene contributions. All MCMC calculations were made using
Matlab scripts (ver. 2014b) using 200,000 iterations, a burn-in phase (initial search
period) of 10,000 and a data thinning of 10.
The spatial extent of ICD in the Lena River Basin was calculated by overlaying
the extent of the drainage basin (from WRIBASIN: Watersheds of the World
published by the World Resources Institute, www.wri.org/publication/watersheds-
world) with the extent of the Yedoma Region (digitized from Romanovsky, 1993) in
an equal area map projection. It was assumed that 30% of the Yedoma Region consists
of intact ICD (following Strauss et al., 2013).

**3    Results**

**3.1    Bulk geochemistry**

The investigated Topsoil-PF and ICD-PF samples are, on a bulk geochemical level, very different. Mean organic carbon contents (as %OC) and total nitrogen content (as %TN) are 25±12 and 1.1±0.67 for Topsoil-PF samples, and 1.6±0.31 and 0.17±0.058 for ICD-PF samples, respectively (Table 1). This gives TOC/TN ratios of 25±8.0 for Topsoil-PF samples and 10±2.6 for ICD-PF samples. Stable carbon isotopic values of Topsoil-PF and ICD-PF samples are -27.8±1.3‰ and -25.7±0.75‰, respectively (Table 1). Radiocarbon ages were unfortunately only available for a few ICD samples, and ranged between 17 and 28 $^{14}$C ka (Table 1).

**3.2    Molecular geochemical composition**

Long-chain $n$-alkanes and $n$-alkanoic acids are abundant in epicuticular waxes and therefore indicative for a source of higher plants (Eglinton and Hamilton, 1967). Concentrations of individual long-chain $n$-alkanes in Topsoil-PF samples ranged from 1 to 340 µg/gOC ($C_{21}$-$C_{33}$; Table 2) with an average chain length of 28±1.6. The sum of high-molecular weight (HMW) $n$-alkanes (>$C_{21}$) for Topsoil-PF samples was $418^{612}_{280}$ µg/gOC (median with interquartile range) and the most abundant $n$-alkanes added up to $214^{494}_{148}$ µg/gOC (sum of $C_{25}$-$C_{27}$-$C_{29}$-$C_{31}$) (Table 4, Fig. 2a). For ICD-PF samples, the individual concentrations of long-chain $n$-alkanes were between 4 and 160 µg/gOC, and the average chain length 27±0.7 (Table 2). The sum of high-molecular weight $n$-alkanes, and most abundant $n$-alkanes were $698^{806}_{630}$ µg/gOC and $347^{405}_{323}$ µg/gOC, respectively (Table 4, Fig. 2a). The carbon preference index (CPI), a molecular ratio indicative for degradation status with values >5 typical for fresher terrestrial material and values approaching 1 typical for more degraded samples (Hedges and Prahl, 1993), showed values for Topsoil-PF samples of 7.3±3.6 (average±standard deviation) and ICD-PF samples of 3.6±0.8 (CPI $C_{23}$-$C_{31}$; Table 4, Fig. 2c). The $C_{25}/(C_{25}+C_{29})$ ratio, indicative for the input of peat moss (*Sphagnum sp.*) material (Vonk and Gustafsson, 2009; *Sphagnum* values 0.72, higher plants 0.07; Nott et al., 2000) was 0.33±0.22 (average±standard deviation) and 0.34±0.05 for Topsoil-PF and ICD-PF samples, respectively (Table 4). Another commonly used *Sphagnum* proxy (Bush and McInerney, 2013), $C_{23}/(C_{23}+C_{29})$, resulted in a sharper contrast between ICD-PF and Topsoil-PF samples (0.39±0.13 and 0.25±0.23, respectively; Fig. 2e and 
[revised manuscript text omitted]

Finally, we realize that the amount of soil and ICD samples analyzed in this study is
limited, and want to point out that the results may change when more data are
analyzed in the near future. Additionally, studies have shown that the $\delta^2H$ signature
of ice within ICD permafrost deposits can range from roughly -150‰ to -260‰
depending on the type of ice (wedge ice vs. pore or texture ice) as well as the period
of formation (different Pleistocene cold stages) (Opel et al., 2017 and references
therein). The source of water (i.e. type of ice) and age of the deposit will therefore
influence the *n*-alkane or *n*-alkanoic acid $\delta^2H$ signal. However, regardless of the
natural variability associated with the processes mentioned above, both ICD and
texture-ice isotopic compositions appear to reflect long-term climate changes (Opel
et al., 2017; Schwamborn et al., 2006; Dereviagin et al., 2013; Porter et al., 2016)
which, likely, were also captured in the *n*-alkane or *n*-alkanoic acid $\delta^2H$ signal.
Unfortunately, we do not have [14]C-ages available for all ICD samples, so cross-
referencing to published stratigraphies in the region is not possible. Coastal
sediments, however, will represent a mixture of material released from different
depths, outcrops, and stratigraphies within the catchment or coast. For source-
apportionment applications, we reason that a growing body of leaf wax $\delta^2H$ end-
member data from the ICD region can overcome the variability issues highlighted
above.

[revised manuscript text omitted]

a vegetation/grass samples, labelled with "G"
b data from Vonk et al., 2013

**Table 2**
Long-chain *n*-alkane concentrations (in μg/gOC) of topsoil Holocene samples (modern vegetation/O-horizon) and Pleistocene ice complex samples.

| | C21 | C22 | C23 | C24 | C25 | C26 | C27 | C28 | C29 | C30 | C31 | C32 | C33 |
|---|---|---|---|---|---|---|---|---|---|---|---|---|---|
| | μg/g OC | | | | | | | | | | | | |
| *Topsoil (modern vegetation and O-horizon samples)* | | | | | | | | | | | | | |
| TS-1 | 44 | 88 | 96 | 45 | 41 | 10 | 45 | 4.4 | 27 | 2.5 | 36 | 1.5 | 7.2 |
| TS-2 | 24 | 15 | 21 | 12 | 40 | 10 | 160 | 10 | 150 | 6.5 | 150 | 3.5 | 17 |
| TS-3 | 2.5 | 2.4 | 5.9 | 2.6 | 13 | 4.7 | 42 | 16 | 74 | 4.7 | 85 | 2.7 | 24 |
| TS-4 | 19 | 3.3 | 7.1 | 2.7 | 27 | 4.5 | 47 | 6.7 | 98 | 9.1 | 150 | 5.7 | 38 |
| TS-5 | 35 | 8.4 | 26 | 9.9 | 38 | 13 | 91 | 18 | 180 | 14 | 230 | 8.1 | 43 |
| TS-6 | 14 | 5.1 | 16 | 5.7 | 19 | 4.0 | 26 | 3.7 | 48 | 5.0 | 120 | 4.0 | 32 |
| TS-7 | 46 | 12 | 18 | 8.8 | 22 | 16 | 61 | 27 | 220 | 23 | 340 | 12 | 48 |
| TS-8G | 4.1 | 1.7 | 18 | 10 | 61 | 16 | 47 | 13 | 30 | 5.3 | 10 | 1.1 | 1.1 |
| TS-9G | 4.7 | 2.6 | 18 | 15 | 45 | 21 | 50 | 16 | 31 | 6.8 | 9.8 | 1.5 | 2.6 |
| *Ice complex deposits* | | | | | | | | | | | | | |
| ICD-1 | 57 | 79 | 100 | 49 | 82 | 23 | 170 | 16 | 137 | 8.5 | 140 | 4.4 | 25 |
| ICD-2 | 55 | 89 | 100 | 70 | 70 | 27 | 75 | 20 | 130 | 12 | 120 | 5.3 | 28 |
| ICD-3 | 40 | 64 | 74 | 31 | 54 | 15 | 79 | 22 | 110 | 10 | 160 | 4.8 | 32 |
| ICD-4 | 60 | 93 | 98 | 47 | 55 | 20 | 84 | 22 | 140 | 12 | 150 | 6.0 | 39 |
| ICD-5 | 46 | 79 | 86 | 56 | 49 | 20 | 55 | 13 | 75 | 7.0 | 100 | 4.7 | 38 |
| ICD-6 | 41 | 73 | 87 | 68 | 62 | 29 | 65 | 20 | 98 | 11 | 120 | 4.9 | 27 |
| ICD-7 | 50 | 83 | 83 | 43 | 41 | 16 | 65 | 17 | 100 | 8.3 | 120 | 4.5 | 42 |
| ICD-8 | 4.2 | 7.3 | 23 | 30 | 55 | 42 | 82 | 38 | 100 | 18 | 110 | 5.0 | 21 |
| ICD-9 | 6.2 | 6.2 | 16 | 11 | 29 | 15 | 51 | 20 | 79 | 9.3 | 85 | 4.1 | 23 |

**Table 3**
Long-chain *n*-alkanoic acids concentrations (in μg/gOC) of topsoil Holocene samples (modern vegetation/O-horizon) and Pleistocene ice complex samples.

| | C16 μg/gOC | C18 | C20 | C21 | C22 | C23 | C24 | C25 | C26 | C27 | C28 | C29 | C30 |
|---|---|---|---|---|---|---|---|---|---|---|---|---|---|
| *Topsoil (modern vegetation and O-horizon samples)* | | | | | | | | | | | | | |
| TS-1 | 511 | 220 | 176 | 80.5 | 539 | 311 | 1100 | 4.95 | 684 | 90.5 | 350 | 32.8 | 58.1 |
| TS-2 | 1740 | 664 | 673 | 235 | 1380 | 496 | 1390 | 543 | 1740 | 409 | 1580 | 113 | 305 |
| TS-3 | 664 | 296 | 480 | 116 | 1020 | 504 | 1710 | 415 | 1550 | 250 | 1060 | 132 | 456 |
| TS-4 | 1140 | 408 | 665 | 235 | 1400 | 431 | 1410 | 425 | 1250 | 242 | 651 | 143 | 455 |
| TS-5 | 513 | 343 | 530 | 133 | 1140 | 359 | 1410 | 1.58 | 896 | 119 | 494 | 67.8 | 224 |
| TS-6 | 1080 | 537 | 418 | 236 | 1420 | 790 | 2670 | 2.82 | 1570 | 127 | 657 | 46.6 | 174 |
| TS-7 | 1420 | 352 | 538 | 281 | 1850 | 722 | 2010 | 651 | 1790 | 642 | 1580 | 730 | 1971 |
| TS-8G | 3640 | 855 | 691 | 44.1 | 609 | 63.5 | 156 | 26.0 | 224 | 0.122 | 99.3 | 9.91 | 28.1 |
| TS-9G | 4600 | 887 | 966 | 53.6 | 815 | 66.7 | 261 | 28.6 | 232 | 11.5 | 124 | 8.10 | 30.2 |
| *Ice complex deposits* | | | | | | | | | | | | | |
| ICD-1 | 1750 | 1600 | 4560 | 1460 | 9460 | 2300 | 8930 | 2020 | 5830 | 1030 | 3660 | 293 | 635 |
| ICD-2 | 10400 | 4030 | 5800 | 2410 | 17100 | 7270 | 18600 | 6610 | 16600 | 5860 | 14800 | 6810 | 18700 |
| ICD-3 | 665 | 554 | 892 | 263 | 2070 | 1060 | 3070 | 646 | 2340 | 272 | 1310 | 133 | 532 |
| ICD-4 | 1400 | 769 | 1030 | 252 | 2040 | 910 | 3120 | 644 | 2440 | 266 | 1160 | 124 | 432 |
| ICD-5 | 426 | 304 | 447 | 126 | 1220 | 511 | 1970 | 70.4 | 1390 | 133 | 712 | 60.7 | 233 |
| ICD-6 | 722 | 539 | 583 | 153 | 1370 | 606 | 2270 | 457 | 1970 | 181 | 1030 | 86.4 | 333 |
| ICD-7 | 446 | 313 | 543 | 158 | 1330 | 562 | 2350 | 401 | 1370 | 154 | 743 | 63.1 | 230 |
| ICD-8 | 920 | 402 | 895 | 108 | 1070 | 294 | 1180 | 184 | 799 | 70.3 | 331 | 34.4 | 100 |
| ICD-9 | 327 | 200 | 559 | 74 | 803 | 229 | 1010 | 2.17 | 718 | 64.9 | 334 | 28.7 | 104 |

**Table 4**

Sum of most abundant long-chain *n*-alkanoic acids and *n*-alkanes (concentrations in µg/gOC), and characteristic ratios of *n*-alkanoic acids and *n*-alkanes of topsoil Holocene (modern vegetation/O-horizon) and Pleistocene ice complex samples.

| | *n*-alkanoic acids | | | | *n*-alkanes | | | | |
|---|---|---|---|---|---|---|---|---|---|
| | ΣHMW[a] (>$C_{22}$) µg/gOC | ΣC$_{22}$-C$_{28}$ (even) µg/gOC | CPI[b] | HMW acids/ HMW alkanes[a] | ΣHMW[a] (>$C_{21}$) µg/gOC | ΣC$_{25}$-C$_{31}$ (odd) µg/gOC | CPI[c] | $C_{23}/$ $(C_{23}+C_{23})$ | $C_{25}/$ $(C_{25}+C_{29})$ |
| *Topsoil (modern vegetation and O-horizon samples)* | | | | | | | | | |
| TS-1 | 3167 | 2670 | 5.8 | 7.1 | 447 | 148 | 2.7 | 0.78 | 0.60 |
| TS-2 | 7958 | 6090 | 3.8 | 13 | 612 | 494 | 11 | 0.12 | 0.21 |
| TS-3 | 7095 | 5340 | 4.1 | 25 | 280 | 214 | 7.2 | 0.07 | 0.15 |
| TS-4 | 6397 | 4700 | 3.7 | 15 | 418 | 323 | 12 | 0.07 | 0.24 |
| TS-5 | 4715 | 3940 | 6.8 | 6.6 | 717 | 543 | 9.1 | 0.12 | 0.17 |
| TS-6 | 7454 | 6310 | 6.0 | 25 | 300 | 211 | 9.9 | 0.25 | 0.28 |
| TS-7 | 11950 | 7230 | 2.9 | 14 | 857 | 647 | 7.8 | 0.08 | 0.09 |
| TS-8G | 1216 | 1090 | 9.5 | 5.6 | 217 | 148 | 3.7 | 0.37 | 0.67 |
| TS-9G | 1577 | 1430 | 11 | 7.1 | 223 | 135 | 2.5 | 0.36 | 0.59 |
| *Mean±stdev* | 5726±3431 | 4310±2190 | 5.9 | 13 | 452±230 | 318±195 | 7.3 | 0.25 | 0.33 |
| *Median and IQR* | $6397^{7454}_{3167}$ | $4700^{2670}_{6092}$ | 2.7 | 7.6 | $418^{621}_{280}$ | $214^{494}_{148}$ | 3.6 | 0.23 | 0.22 |
| *Ice complex deposits* | | | | | | | | | |
| ICD-1 | 34854 | 27883 | 4.1 | 39 | 893 | 530 | 4.9 | 0.43 | 0.38 |
| ICD-2 | 112356 | 67078 | 2.8 | 140 | 806 | 398 | 3.0 | 0.44 | 0.35 |
| ICD-3 | 11430 | 8791 | 4.1 | 16 | 698 | 405 | 4.6 | 0.40 | 0.33 |
| ICD-4 | 11145 | 8768 | 4.4 | 14 | 825 | 428 | 3.8 | 0.42 | 0.29 |
| ICD-5 | 6293 | 5285 | 6.5 | 10 | 630 | 280 | 2.9 | 0.54 | 0.40 |
| ICD-6 | 8293 | 6629 | 4.9 | 12 | 708 | 347 | 2.7 | 0.47 | 0.39 |
| ICD-7 | 7196 | 5787 | 4.7 | 11 | 671 | 323 | 3.5 | 0.45 | 0.29 |
| ICD-8 | 4063 | 3380 | 5.5 | 7.6 | 533 | 344 | 2.7 | 0.19 | 0.35 |
| ICD-9 | 3295 | 2867 | 8.3 | 9.3 | 355 | 244 | 4.3 | 0.17 | 0.27 |
| *Mean±stdev* | 22103±35150 | 15160±20800 | 5.0 | 29 | 680±163 | 367±85 | 3.6 | 0.39 | 0.34 |
| *Median and IQR* | $8290^{11430}_{6290}$ | $6630^{8790}_{5285}$ | 1.6 | 43 | $698^{806}_{630}$ | $347^{405}_{323}$ | 0.8 | 0.13 | 0.05 |

a HMW; high-molecular weight b CPI; carbon preference index for chain lengths $C_{22}$-$C_{28}$, for calculation see caption of Fig. 2.

c CPI; carbon preference index for chain lengths $C_{23}$-$C_{31}$, for calculation see caption of Fig. 2.

**Formatted Table** ... [1]
**Formatted Table** ... [2]
**Formatted Table** ... [3]
**Formatted Table** ... [5]

**Table 5**

$\delta^2$H signatures (in ‰) of *n*-alkanoic acids and *n*-alkanes of topsoil Holocene (modern vegetation/O-horizon) and Pleistocene ice complex samples.

| | *n*-alkanoic acids | | | | | | | *n*-alkanes | | | |
|---|---|---|---|---|---|---|---|---|---|---|---|
| | C16 | C18 | C20 | C22 | C24 | C26 | C28 | C25 | C27 | C29 | C31 |
| **Topsoil (modern vegetation and O-horizon samples)** | | | | | | | | | | | |
| TS-1 | -162 | -180 | | -119 | -178 | -203 | -197 | -168 | -240 | -236 | -244 |
| TS-2 | -188 | -192 | | -211 | -222 | -232 | -225 | -196 | -237 | -251 | -239 |
| TS-3 | | | | -126 | -203 | -218 | -225 | | -125 | -234 | -259 |
| TS-4 | -171 | -213 | | -180 | -196 | -210 | -206 | | -182 | -245 | -243 |
| TS-5 | | -235 | -185 | -253 | -257 | -264 | -244 | -164 | -240 | -266 | -273 |
| TS-6 | -189 | -222 | | -214 | -235 | -236 | -224 | -203 | -221 | -258 | -282 |
| TS-7 | -184 | | -190 | -218 | -227 | -225 | -220 | -199 | -234 | -259 | -250 |
| TS-8G | -258 | -246 | -253 | -256 | -285 | -286 | | -253 | -236 | -234 | -224 |
| TS-9G | -237 | -244 | -251 | -248 | -245 | -248 | | -227 | -223 | -234 | -209 |
| *Mean* | *-199* | *-219* | *-220* | *-203* | *-228* | *-236* | *-220* | *-201* | *-215* | *-246* | *-247* |
| *St.dev* | 35 | 25 | 37 | 52 | 33 | 27 | 15 | *32* | *39* | *13* | *23* |
| **Ice complex deposits** | | | | | | | | | | | |
| ICD-1 | -194 | -227 | -243 | -252 | -245 | -241 | -232 | -237 | -257 | -268 | -265 |
| ICD-2 | | | -231 | -264 | -271 | -280 | -271 | -217 | -266 | -283 | -297 |
| ICD-3 | | | -249 | -262 | -278 | -264 | | -254 | -279 | -283 | -307 |
| ICD-4 | | | -209 | -252 | -266 | -277 | -254 | -243 | -261 | -285 | -305 |
| ICD-5 | | | -169 | -260 | -275 | -288 | -273 | -189 | -245 | -269 | -283 |
| ICD-6 | -211 | -216 | -252 | -266 | -274 | -294 | -285 | -192 | -254 | -281 | -296 |
| ICD-7 | | | -191 | -263 | -277 | -287 | -273 | -210 | -279 | -295 | -309 |
| ICD-8 | -244 | -256 | -277 | -277 | -277 | -293 | -280 | -195 | -221 | -263 | -298 |
| ICD-9 | -228 | -229 | -261 | -265 | -262 | -262 | -267 | -251 | -270 | -295 | -313 |
| *Mean* | *-219* | *-232* | *-229* | *-261* | *-268* | *-278* | *-267* | *-221* | *-259* | *-280* | *-297* |
| *St.dev* | 21 | 17 | 37 | 8.6 | 10 | 17 | 16 | *26* | *18* | *12* | *15* |

**Table 6**

Location, sampling depth and isotopic values of samples along a surface sediment transect in the Laptev Sea (data from Bröder et al., 2016b), with percentage topsoil (TS) and ice complex deposit (ICD) OC contributions to the samples based on source-apportionment calculations with $\delta^2$H leaf wax end-members versus $\delta^{13}$C-$\Delta^{14}$C end-members (end-member values are described in the text).

| | | | | Sample values | | | | | | Source contributions | | | |
|---|---|---|---|---|---|---|---|---|---|---|---|---|---|
| ID[a] | Lat | Long | Depth | $C_{27}$ | $C_{29}$ | $C_{31}$ | $C_{27\text{-}29\text{-}31}$[b] | $\delta^{13}$C | $\Delta^{14}$C | TS | ICD | TS[c] | ICD[c] |
| | N | °E | m | ‰ | ‰ | ‰ | ‰ | ‰ | ‰ | using $\delta^2$H | | using $\delta^{13}$C-$\Delta^{14}$C | |
| TB-46 | 72.700 | 130.180 | 6 | -236.2 | -237.4 | -230.4 | -235.0 | -26.5 | -436 | 89% | 11% | 63% (63%) | 37% (37%) |
| YS-9 | 73.366 | 129.997 | 23 | -233.7 | -231.0 | -227.8 | -231.1 | -26.1 | -415 | 91% | 8.9% | 63% (65%) | 37% (35%) |
| YS-6 | 74.724 | 130.016 | 32 | -234.2 | -241.0 | -235.4 | -236.8 | -25.6 | -465 | 86% | 14% | 51% (59%) | 49% (41%) |
| SW-24 | 75.599 | 129.558 | 46 | -229.3 | -236.5 | -243.5 | -236.4 | -24.8 | -284 | 87% | 13% | 70% (72%) | 30% (28%) |
| SW-23 | 76.171 | 129.333 | 56 | -219.9 | -243.3 | -243.3 | -236.0 | -25.0 | -333 | 83% | 17% | 65% (70%) | 35% (30%) |
| SW-06 | 77.142 | 127.378 | 92 | -219.5 | -237.0 | -241.4 | -233.2 | -23.2 | -364 | 87% | 13% | 39% (53%) | 61% (47%) |
| SW-03 | 78.238 | 126.150 | 2601 | -221.1 | -238.0 | -247.7 | -235.9 | -22.6 | -426 | 85% | 15% | 23% (42%) | 77% (58%) |
| SW-01 | 78.942 | 125.243 | 3146 | -223.8 | -241.8 | -246.0 | -238.0 | -22.3 | -418 | 83% | 17% | 21% (42%) | 79% (58%) |

a Location, depth and bulk carbon isotope data from Bröder et al. (2016b)

b weighted average based on individual concentrations c numbers in brackets are source contributions using the $\delta^{13}$C-$\Delta^{14}$C approach but with additional corrections for cross-shelf lateral transport time of topsoil OC (similar as in Bröder et al., 2016a); we applied linear aging along the transect based on the distance from the coast, with a maximum aging of 5000 years for station SW-01.

| Page 32: [1] Formatted Table | Jorien | 5/25/17 9:04:00 AM |
|---|---|---|

Formatted Table

| Page 32: [2] Formatted Table | Jorien | 5/25/17 9:05:00 AM |
|---|---|---|

Formatted Table

| Page 32: [3] Formatted Table | Jorien | 6/3/17 3:00:00 PM |
|---|---|---|

Formatted Table

| Page 32: [4] Deleted | Jorien | 6/3/17 2:59:00 PM |
|---|---|---|

*St.dev*

| Page 32: [4] Deleted | Jorien | 6/3/17 2:59:00 PM |
|---|---|---|

*St.dev*

| Page 32: [4] Deleted | Jorien | 6/3/17 2:59:00 PM |
|---|---|---|

*St.dev*

| Page 32: [5] Formatted Table | Jorien | 6/3/17 3:11:00 PM |
|---|---|---|

Formatted Table